



# Experiments with $CO_2$-in-air reference gases in high-pressure aluminum cylinders

Michael F. Schibig[1], Duane Kitzis[1,2] and Pieter P. Tans[1]

[1] Global Monitoring Division, Earth System Research Laboratory, National Oceanic and Atmospheric Administration, Boulder, Colorado, 80305, USA
[2] Cooperative Institute for Research in Environmental Sciences, University of Colorado, Boulder, Colorado 80309, USA

*Correspondence to*: pieter.tans@noaa.gov

**Abstract**

Long term monitoring of carbon dioxide ($CO_2$) in the atmosphere is key for a better understanding of the processes involved in the carbon cycle that have a major impact on further climate change. Keeping track of large-scale emissions and removals ("sources and sinks") of $CO_2$ requires very accurate measurements. They all have to be calibrated very carefully and have to be traceable to a common scale, the WMO $CO_2$ X2007 scale, which is maintained by NOAA/ESRL (Oceanic and Atmospheric Administration/Earth System Research Laboratory) in Boulder, CO, USA. The international WMO/GAW (World Meteorological Organization/Global Atmosphere Watch) program sets as compatibility goals for the required agreement between different methods and laboratories $\pm 0.1$ µmol mol$^{-1}$ for the northern hemisphere and $\pm 0.05$ µmol mol$^{-1}$ for the southern hemisphere. The reference gas mixtures used to pass down and distribute the scale are stored in high pressure aluminum cylinders. It is crucial that the standards remain stable during their entire time of use. In this study we found that during low flow conditions (0.3 l min$^{-1}$) the tested vertically positioned aluminum cylinders always showed similar $CO_2$ enrichment of $0.090 \pm 0.009$ µmol mol$^{-1}$ as the cylinder was emptied from about 140 to 1 bar above atmosphere, following Langmuir's adsorption/desorption model. When decanted at a higher rate of 5.0 l min$^{-1}$ the enrichment becomes $0.22 \pm 0.05$ µmol mol$^{-1}$ for the same pressure drop. The higher enrichment is related to thermal diffusion and fractionation effects in the cylinder, which were also dependent on the cylinder's orientation and could even turn negative. However, the low amount of $CO_2$ adsorbed on the cylinder wall as well as the fact that the main increase happens at low pressure lead to the conclusion that aluminum cylinders are suitable to store ambient $CO_2$-in-dry-air mixtures provided they are not used below 20 bar. In case they are used in high flow experiments that involve significant cylinder temperature changes, special attention has to be paid to possible fractionation effects.

## 1. Introduction

The amount of the emissions in combination with the radiative forcing makes carbon dioxide ($CO_2$) the most important anthropogenic greenhouse gas (GHG) (IPCC, 2013; Hofmann et al., 2006). $CO_2$ exchanges rapidly between the atmosphere, oceans and terrestrial biosphere (the "fast exchange"), and very slowly with carbonate rocks. The current combustion of coal, oil, and natural gas constitutes a large scale transformation of fossilized





organic matter to $CO_2$ gas that is now overwhelming natural exchange processes. The $CO_2$ emissions are practically irreversible; removal from the atmosphere and oceans by natural sedimentation and erosion will take thousands of years. The fast exchange implies that not only does $CO_2$ influence climate, but the oceans as well as the terrestrial biosphere can gain or lose carbon as climate change unfolds, which is often called the "carbon-climate feedback".

This feedback constitutes a major uncertainty for climate projections. We need to create an accurate record of changing sources/sinks to the atmosphere in order to diagnose and quantify these feedbacks as they occur.

Downwind of a source region atmospheric $CO_2$ is enhanced relative to upwind. However such enhancements/depletions due to regional sources/sinks are typically very small on regional to continental scales, so that long term monitoring with very accurate measurements is necessary. Small systematic errors between

measurement stations can lead to mis-assignment of sources or sinks, noisy measurements might obscure interesting signals that could help to identify processes and calculate their contribution to the carbon cycle (e.g. Masarie et al., 2011). High quality measurements start with careful calibrations, preferably traceable to the International System of Units (SI), or if not possible as in the case of isotopic ratios, to an artifact chosen by convention (e.g., the kilogram, or VPDB for $^{13}C/^{12}C$). In the case of GHGs traceability is maintained by the use of a unique hierarchy of $CO_2$-in-

(dry)-air mixtures (and similarly for $CH_4$, $N_2O$) in high pressure cylinders, starting from the primary standards (with link to SI) to secondaries and tertiaries, all with known $CO_2$ mole fraction derived from the higher level, ultimately calibrating the instrument making air measurements. Careful calibration procedures make the result independent of which instrument or method is used. The resulting data stand on their own feet; they do not depend on models or a-priori estimates and assumptions, and are true within a known uncertainty range.

The World Meteorological Organization (WMO) coordinates GHG measurements around the world, through its Global Atmosphere Watch Program (GAW), and during biannual meetings of the international participating laboratories ("the community") goals have been set for the level of compatibility between different stations. The community recommends the WMO $CO_2$ X2007 scale (WMO, 2014), and they defined a compatibility goal of $\pm$ 0.1 µmol mol$^{-1}$ (1 standard deviation) for $CO_2$ datasets of the northern hemisphere (WMO, 2014, 2012, 2011,

2007, 2006; Zellweger et al., 2016). For the southern hemisphere this number is even lower at $\pm$ 0.05 µmol mol$^{-1}$ because smaller source intensities, due to the large proportion of ocean surface, give rise to smaller spatial gradients than in the northern hemisphere. The WMO $CO_2$ X2007 scale is embodied in 15 primary standards which are measured once every two years on a manometric system that provides SI values by NOAA/ESRL (National Oceanic and Atmospheric Administration/Earth System Research Laboratory) in Boulder, USA (Zhao and Tans, 2006; Zhao

et al., 1997). The primary standards are used to transfer the calibration scale to secondary and subsequently to tertiary standards. The tertiary standards are sent to the different laboratories around the world to calibrate their $CO_2$ measurements. To meet the WMO's accuracy goal of 0.1 (or 0.05) µmol mol$^{-1}$, it is crucial that the standards remain stable during their entire time of use, and/or that they are re-calibrated at reasonable intervals, and that appropriate laboratory practices are being followed. The latter are included in the biannual WMO reports as "Expert Group

Recommendations". At field stations (Schibig et al., 2015) but also in laboratory experiments (Langenfelds et al., 2005; Leuenberger et al., 2015; Miller et al., 2015), standard gases typically show some $CO_2$ enrichment with decreasing pressure. Those studies attributed the $CO_2$ enrichment to different effects such as Langmuir monolayer





adsorption/desorption, gravimetric fractionation, thermal fractionation or Rayleigh distillation related effects for example. Evaluating 10 years of calibration tank measurements, Keeling et al. (2007) found a downward drift in their aluminum calibration tanks relative to steel, which they attributed to surface conditioning.

In this study the following hypotheses were tested, i) the $CO_2$ increase with decreasing pressure is different for each individual cylinder, ii) the $CO_2$ enrichment follows the Langmuir monolayer adsorption/desorption model and iii) the stability of the $CO_2$ mole fraction is better in SGS (Superior Gas Stability®, Luxfer, USA) cylinders. To check the first hypothesis, eight cylinders were repeatedly filled and decanted and the $CO_2$ enrichment of the individual measurements was compared. The second hypothesis was investigated by decanting the cylinders at different flow rates. At low flow the temperature changes due to the decreasing pressure are negligible, whereas at a high flow setting the fast pressure decrease induces cooling and substantial temperature gradients in the cylinder. If only adsorption and desorption effects are at work, the $CO_2$ enrichment can be expected to be the same as with the low flow experiments, unless the wall equilibration times are long (at least several hours) such that during high flow experiments the walls do not equilibrate as during low flow experiments. In that case one could expect to see a smaller wall effect. Additionally, the cylinders were positioned in different orientations, which again shouldn't have any influence on the measured $CO_2$ mole fraction of the outflowing gas if only adsorption/desorption effects are involved. Furthermore, heating bands were used to alter the temperature of the cylinder wall to learn more about potential temperature issues. To check the third hypothesis, two SGS cylinders were added to the set but used exactly the same way as the ordinary cylinders. If their surface treatment is beneficial to the $CO_2$ stability, the experiments with SGS cylinders should stand out clearly.

## 2. Methods

The $CO_2$ measurement system was based on a customized replacement unit of NOAA's tall tower network (Andrews et al., 2014). To measure $CO_2$ mole fractions a nondispersive infrared gas analyzer (LI-7000, LI-COR, USA) was used. In March 2017, the original $CO_2$ analyzer stopped working and had to be replaced by a spare analyzer of the same make and model. The system was controlled by CR-1000 measurement and control data logger (Campbell Scientific, USA), a Windows laptop was used to communicate with the CR-1000 and to store the data. The pressure regulators were equipped with digital pressure readers on the high pressure side (EW-68075-10, Cole-Parmer Instrument Company, USA), the pressure is measured relative to atmospheric pressure. To avoid complete drainage, the sample cylinders were excluded from the measurement sequence as soon as the primary pressure dropped below a preset threshold (1 bar in low flow experiments, 1.5 bar in high flow experiments). The calibrations and measurements were done in a cyclic sequence defined in the control program. The cycles were divided in blocks of 5 minutes each, during which gas from a single cylinder (sample, calibration or target gas) was measured. The data was read every 5 seconds, but data reported to the log file were 30 s averages yielding 10 values per block. In the very beginning of each cycle, a full calibration with all four calibration gases (C1, C2, C3, C4) as well as a target gas measurement were done. Then the program cycled through all samples several times, before C2 was measured again to catch short term drifts in the measurement system (Fig. 1). For quality control, the target gas block was also measured in between the samples. An additional full calibration was measured at the end of each experiment. The



values reported by the LI-7000 are Δ-signals which are basically the difference between the sample and the continuously at 10 ml min$^{-1}$ flowing reference cell signals.

To calculate the $CO_2$ mole fractions of the sample and target measurements, the raw Δ-signals of the calibration gas measurements were interpolated over time and together with the assigned values of the calibration gases a quadratic

calibration function was calculated for each individual sample or target measurement. To guarantee a proper flushing in between different gases and to avoid memory and mixing effects, only the last 2 minutes of each 5 minute block were averaged into one $CO_2$ mole fraction value and used for further calculations. Since we were only dealing with dry cylinder gases ($H_2O < 1$ µmol mol$^{-1}$), the drying unit was bypassed and no water correction was applied. The measurements were done in two different flow settings, a low flow setting with 0.3 l min$^{-1}$ drain and a

high flow setting with 5.0 l min$^{-1}$. The different settings required some adjustments of the hardware and calibration procedure, which will be explained in the following sections.

### 2.1 Low Flow measurements

In the low flow setting the cylinders were hooked to a VICI multiport valve, which was used to switch between the different calibration gases and the sample gas cylinders. Until 8.11.2016, it was a 10-port valve (EMT2SD10MWE,

Valco Instruments Co. Inc., USA), where ports 1 to 5 were used for the sample gases and ports 6 to 10 were connected to C1, C2, C2, C4 and the target gas. The valve was later upgraded to a 16-port valve (EMT2SD16MWE, Valco Instruments Co. Inc., USA), which allowed measuring all sample cylinders in one run. With the new 16-port valve ports 1 to 8 were used for the samples, ports 9 to 13 were used for the calibration and target gases. The secondary pressure for the calibration as well as the target and sample gas was set to about 1 bar. To allow a

constant flow out of the cylinder throughout the whole measurement, solenoids were used in each sample line to open by-pass lines for the cylinders that weren't currently measured. The by-pass lines led to needle valves, where the flow was adjusted to 0.3 l min$^{-1}$ for each sample cylinder individually. The by-pass line of the cylinder currently measured remained closed. This ensured that all the gas flowed through the analyzer, kept the flow rate coming out of the cylinder stable and avoided potential fractionation at the tee unions (Fig. 2 a) due to pressure and/or

temperature gradients between the arm and the runs of the tee unions. In the low flow measurements with multiple samples, all samples were subsequently measured twice before a block of C2 was measured, and this was repeated three times, before another full calibration was done. In case all eight sample lines were used, one cycle took 275 min, a whole run lasted about 9 to 10 days. In the low flow measurements, the target gas block was added every 1000$^{th}$ minute, which resulted in about 12 additional target gas measurements in one complete run.

### 2.2 High flow measurements

In the high flow setting, the gas was drained out of the cylinder at 5.0 l min$^{-1}$, which is why a cylinder lasts only about 12 to 13 hours. Only one sample tank was measured per run, otherwise too much detail might be lost, especially towards the end of the experiment (Fig. 2 b). The calibration sequence was similar to the low flow measurements with some minor changes. In the beginning of every cycle there was again a complete calibration

sequence. After the calibration, the sample block was repeated 10 times followed by one block of C2 and again 10





blocks of sample gas that completed the cycle. A target gas block was added every 150[th] minute, which added about 4 target gas measurements. In order to catch as much sample measurements within the last few bars of the sample cylinders lifetime, some additional conditions related to the sample pressure were added to the measurement sequence: i) no full calibration below 35 bar sample pressure, ii) no target gas measurement below 15 bar and iii) no

C2 below 8 bar. The initial calibration in the low flow measurements sometimes showed noisy measurements, most probably due to run-in effects of the whole system. While four hours in the low flow measurements corresponds only to a fraction of the whole run, it would be about a third of a high flow run. To avoid these effects a two hour flush cycle with gas similar to the sample gases was added prior to the first calibration measurement.

### 2.3 High flow inlet system

Originally the measurement system was designed to operate with sample gas flows of about 0.2-0.3 l min$^{-1}$. To achieve a flow of 5.0 l min$^{-1}$ out of a sample cylinder, 4.7-4.8 l min$^{-1}$ had to be bypassed in a non-fractionating manner. To do so an inlet system similar to an open-split design was built (Fig. 3). The gas enters the inlet system on one end at a flow rate of 5.0 l min$^{-1}$ and flows through a 0.5 inch stainless steel union and then through a 0.5 inch synflex tube. In the center of 0.5 inch tube union, a $^1/_8$ inch stainless steel tube takes an aliquot or air to the

measurement system, the rest leaves at the other end through the exhaust. The length of the outer tube is 0.25 m, which in combination with the high flow is sufficient to avoid back diffusion of outside air through the open split. Because the ambient pressure is too low for the pressure controller to maintain the set pressure of 1030 mbar, the secondary pressure of the pressure regulator of the sample gases was increased to 1.5 bar and a needle valve was used to provide a small backpressure. The needle valve increased or decreased the resistance of the exhaust to the

lab and thereby the ratio of the flows. Measurements of the same cylinder connected to the high flow inlet and the normal flow inlet resulted in the same $CO_2$ mole fraction proving that the sampled small flow is not fractionated from the large bypass flow.

### 2.4 Temperature measurement

Thermistors were used during the high flow experiments to measure the temperature development of the cylinder

surface and on the pressure regulator. The used thermistors were negative temperature coefficient (NTC) sensors (PR103J2, U.S. Sensor Corp., USA) with an accuracy guaranteed by the manufacturer of ± 0.05 °C. Testing the thermistors against a NIST (National Institute of Standards and Technology, USA) calibrated platinum probe showed that they perform even slightly better. The voltages of the thermistors were measured by a Keysight 34901A Data acquisition/switch unit with a 34908A 40-channel multiplexer module (Keysight Technologies, USA) and

logged on a laptop PC. During high flow measurements, the temperature was read once a minute, during the one low flow experiment with temperature measurement once every five minutes. The voltage was converted into temperature values by using the Steinhart–Hart equation (Steinhart and Hart, 1968):

$$\frac{1}{T} = A + B \cdot \ln R + C \cdot (\ln R)^3 \qquad (1)$$





where T is the Temperature, A, B and C are the Steinhart-Hart coefficients provided by the manufacturer of the thermistors and R is the measured resistance. To fix the thermistors to the regulator (T1–T6), the cylinder valve (T7–T8) as well as the cylinder (T9–T18) and insulate them from influences from room air, small pieces of rubber foam and duct tape were used. To detect potential biases in the temperature measurement, two thermistors were

fixed at similar positions opposing each other, and the thermistors (T19–T25) not attached to the cylinder were bundled and used as background measurement (Fig. 4 a and b).

## 2.5 Heating

To learn more about the involved processes, heat was applied in some measurements, steadily over a longer time period or as a single burst. To do so, small heating bands (Minco, USA) with an overall heating power of 110 watts

were attached to the cylinder by using aluminum foil tape. Eight bands were equally distributed in pairs on four levels along the cylinder, the ninth band was attached to the bottom of the cylinder (Fig. 4 a). The heaters were switched on and off by the measurement sequence, the end temperature was set to 30 °C and regulated by a control unit with a thermostat. For safety reasons the cylinders were wrapped in a thin fire proof glass wool mat during these experiments, the insulating effect of the glass wool mat should be negligible.

In case the heat was applied over a longer time period, the heating started as soon as the cylinder reached 30 bar. Heating the whole cylinder to 30 °C took about 1 hour. To avoid losing a lot of gas while the set temperature was not reached, the sample gas flow was shut off during this period and the idle time was used to measure all calibration gases. During the remaining time, flush gas was measured in order to keep the system in steady state.

For the heat bursts, the setup was slightly different. The heating started when the cylinder pressure reached 50 bar

and lasted until the thermostat measured 30 °C, usually at about 40 bar. During the heating, the measurement cycle continued without any further changes.

## 2.6 Sample cylinders

To measure the $CO_2$ development over the lifetime of a cylinder eight cylinders were repeatedly filled to about 130 bar at Niwot Ridge Station, CO, USA. The filling was done the exact same way as for the standard cylinders NOAA

fills to be used as calibration gas tanks (Kitzis, 2017). Six of the eight cylinders were Luxfer L6X® aluminum cylinders, which are the same type NOAA uses for the $CO_2$ standards, two were Luxfer L6X® SGS (Superior Gas Stability) aluminum cylinders (Table 1). Cylinder CB11876 was excluded from the set and was replaced with CB11941 after the first run, because the cylinder valve's sealing surface got scratched badly. The 29.5 l cylinders were fitted with Rotarex Series D200 brass packless valves (Rotarex, Luxembourg) to be used with two stage

pressure reducers (Scott Specialty model 51-14B-590) and chromed brass CGA connection. The regulators were connected to the measurement system by Quick-Connects (SS-QC4-B-4PM and SS-QC4-D-400 for the samples and SS-QM2-B-2PM and SS-QM2-D-200 for the standards, respectively, Swagelok, USA) and 1/8" stainless steel tubing (Swagelok, USA).



**2.7 CO$_2$ enrichment estimates**

Each low flow measurement run of every cylinder was used to fit individually a function based on the Langmuir adsorption/desorption model (Langmuir, 1918, 1916; Leuenberger et al., 2015):

$$CO_{2,meas} = CO_{2,ad} \cdot \left( \frac{K \cdot (P - P_0)}{1 + K \cdot P} + (1 + K \cdot P_0) \cdot \ln \left( \frac{P_0 \cdot (1 + K \cdot P)}{P \cdot (1 + K \cdot P_0)} \right) - 1 \right) + CO_{2,ini} \qquad (2)$$

where CO$_{2,meas}$ corresponds to the measured CO$_2$ mole fraction, CO$_{2,ad}$ stands for the CO$_2$ molecules adsorbed by the cylinder wall expressed as a mole fraction, CO$_{2,ini}$ is the CO$_2$ mole fraction at the start pressure P$_0$, P is the actual pressure in bar and K is the ratio of the adsorption and desorption rate constants and has the units bar$^{-1}$ (see Leuenberger et al. (2015) for more information). To find CO$_{2,ad}$ and CO$_{2,ini}$, a R script using the Nonlinear Least Square fitting algorithm (nls) was used. Because the CO$_2$ enrichment in aluminum cylinders was small, the fit seems to be relatively insensitive to K. Therefore the algorithm wasn't able to find K values with a high confidence level and the output corresponded mostly to the lower boundary, even when it was set to 0, meaning no exchange between the cylinder wall and the gas. To find a value for K nonetheless, a different approach was used. Given that the residuals of a good fit are normally distributed, K can be found by fitting the adsorption/desorption equation but with a fixed K value, starting at a value close to 0 and increase it step wise, until the residuals are not normally distributed anymore. To improve the sensitivity, only CO$_2$ measurements below 30 bar were taken into account, where the CO$_2$ increase is more pronounced. This was done for ten different low flow cylinder measurements. The resulting K values were averaged and the standard deviation was calculated. To make sure the residuals of all fits stay well within the normally distributed range, the K value was considered to be difference "average - standard deviation", which resulted in 0.002 bar$^{-1}$. To make sure, the residuals are normally distributed, K was set in the nls algorithm to 0.001 bar$^{-1}$.

We also took a different approach to find values for the adsorbed CO$_2$ and the exchange rate K starting from a slightly altered Langmuir's adsorption desorption model (Langmuir, 1918, 1916):

$$\theta = \frac{K \rho_x}{1 + K \rho_x} \qquad (3)$$

with θ the fraction of available wall space that is occupied (dimensionless), K again the ratio of the adsorption and desorption rate constants here in the units m$^3$ mol$^{-1}$ and $\rho_x$ the average amount density (mol m$^{-3}$) of CO$_2$ in the gas phase. In this approach θ is a function of CO$_2$ only, not of total gas pressure. With the trace gas mole fraction X and the average amount density $\rho_a$ of air molecules, $\rho_x$ can be written as the product of two independent variables:

$$\rho_x = X \cdot \rho_a \qquad (4)$$

Assuming the ideal gas law with P the pressure, V the volume, R the gas constant, T the temperature, and N$_a$ the total amount of air (moles) in the gas phase, we also have





$$\rho_a = \frac{N_a}{V} = \frac{P}{R \cdot T} \quad and \quad \rho_x = \frac{X \cdot P}{R \cdot T} \tag{5}$$

The amount of molecules adsorbed to the wall $n_{ad}$ can be expressed as:

$$n_{ad} = \theta \cdot a \tag{6}$$

in which a is the available wall space (amount of sites, expressed in moles), a number that we do not know, so that we have for the total amount of trace gas, in the gas phase and on the wall divided by the volume:

$$\frac{Total\ trace\ gas}{V} = X\rho_a + \theta \cdot \frac{a}{V} \tag{7}$$

By draining some air ($d\rho_a$, which is negative) with its current mole fraction X out of a cylinder, a certain amount of
5    trace gas will be removed ($Xd\rho_a$) and the partitioning between the gas phase and the wall will change. If we also assume that the relevant quantities are uniform inside the cylinder, the corresponding change of trace gas per volume can be written as

$$\frac{d(Total\ trace\ gas)}{V} = Xd\rho_a = d(X\rho_a) + d\left(\frac{a}{V} \cdot \theta\right) = \rho_a dX + Xd\rho_a + d\left(\frac{a}{V} \cdot \frac{K\rho_x}{1 + K\rho_x}\right) \tag{8}$$

which we rearranged into the following equation

$$\frac{dX}{X} = -\frac{d\rho_a}{\rho_a} \cdot \frac{\frac{a}{V} \cdot K}{(1 + K\rho_x)^2 + \frac{a}{V} \cdot K} \tag{9}$$

The gas pressure (and the amount density) varies over a large range from 150 to 1 bar, whereas the quotient
10    $\frac{\frac{a}{V}K}{(1+K\rho_x)^2+\frac{a}{V}\cdot K}$ varies only little. Therefore we can integrate Eq. (9) numerically in successive steps from 150 bar to 1 bar, as follows:

$$X_i = X_{i-1} \left(\frac{\rho_{a,i}}{\rho_{a,i-1}}\right)^{\frac{-\frac{a}{V} \cdot K}{(1 + K\rho_{x,i-1})^2 + \frac{a}{V} \cdot K}} \tag{10}$$

Assuming the ideal gas law, it could also be rewritten as

$$X_i = X_{i-1} \left(\frac{P_i}{P_{i-1}}\right)^{\frac{-\frac{a}{V} \cdot K}{(1 + K\rho_{x,i-1})^2 + \frac{a}{V} \cdot K}} \tag{11}$$





with P being the pressure (bar). However, also with this different approach, it wasn't possible to determine K and a independently. There is not enough information in the data at the low observed enrichments. A range of solutions, in which there is a tight anti-correlation between K and a, can reproduce the enrichment at 1 bar. However, with higher K values the enrichment effect becomes more and more concentrated at low pressures, so that at some point the

observed shape of the observations can not be met. Therefore K and the corresponding coverage factor θ (at 150 bar) have to be low.

Additionally to the Langmuir adsorption/desorption model, a Rayleigh distillation function (Langenfelds et al., 2005; Matsubaya and Matsuo, 1982; Rayleigh, 1902) was fitted to the data as well, it had the form:

$$\frac{X}{X_0} = \left(\frac{P}{P_0}\right)^{\alpha-1} \tag{12}$$

where X corresponds to the measured $CO_2$ mole fraction, $X_0$ to the initial $CO_2$ mole fraction, P and $P_0$ correspond to

the actual and initial pressure, respectively, and α is the fractionation factor for the gas leaving the cylinder. The outflowing gas is depleted in $CO_2$ if α < 1 and vice versa. If only low flow experiments are considered, it is not possible to distinguish between Langmuir adsorption/desorption effects and Rayleigh fractionation, both functions give equally reasonable results, which is one reason the high flow experiments were needed (see hypothesis ii in the introduction). To each of the high flow measurements a fit based on the Langmuir adsorption/desorption as well as a

fit based on the combination of the Langmuir and Rayleigh distillation function were calculated. The Langmuir fit was again calculated with K fixed at 0.001 bar$^{-1}$, $CO_{2,ad}$ and $CO_{2,init}$ were estimated by a non-linear least squares (nls) algorithm in a R script. In case of the combination, the Langmuir part was calculated with fixed coefficients that correspond to the averaged coefficients of the low flow experiments. The coefficients $X_0$ and α of the Rayleigh distillation term were again determined using R's nls algorithm using the following equation:

$$CO_{2,meas} = \overline{CO_{2,ad,lf}} \cdot \left(\frac{K \cdot (P - P_0)}{1 + K \cdot P} + (1 + K \cdot P_0) \cdot \ln\left(\frac{P_0 \cdot (1 + K \cdot P)}{P \cdot (1 + K \cdot P_0)}\right) - 1\right) + X_0 \cdot \left(\frac{P}{P_0}\right)^{(\alpha-1)} \tag{13}$$

Where $CO_{2,ad,lf}$ is the average $CO_{2,ad}$ coefficient of the low flow experiments, K is again the ratio of the adsorption and desorption rate constants (fixed at 0.001 bar$^{-1}$), P is the actual pressure, $P_0$ is the initial pressure, $X_0$ corresponds to the $CO_2$ mole fraction before the enrichment and α is the fractionation factor, close to one. To test whether the enrichment follows a Rayleigh fractionation, $\ln(X/X_0)$ can be plotted against $-\ln(P/P_0)$ after the data has been corrected for Langmuir adsorption/desorption effects. If there is Rayleigh fractionation, the points should line up,

following a line with a slope of 1-α.

## 3. Results

No general filtering was applied to the measured data. However, in the beginning of these experiments, some measurements showed run-in effects between the first two calibration points, most probably due to an insufficiently flushed reference line due to the very small flow. These measurements were excluded from any further calculations.



The run-in effects vanished mostly when an additional flush gas cylinder with a 2 hour flushing sequence was added to the measurements.

### 3.1 Accuracy and precision

To estimate the accuracy and repeatability of the system, differences of target $CO_{2,measured}$ minus target $CO_{2,assigned}$ were calculated for each target gas measurement. The differences show a normal distribution with a small positive bias of $0.02 \pm 0.02$ μmol mol$^{-1}$. There seems to be a difference in the target gas measurements before and after the end of March 2017 (Fig. 5 a). Between these two periods C2 had to be changed because it reached the end of its lifetime. A few days later the $CO_2$ analyzer had to be changed as well due to a malfunction. Before that period, the average of the target gas differences was $0.01 \pm 0.01$ μmol mol$^{-1}$. After the C2 and the $CO_2$ analyzer were replaced, it became $0.03 \pm 0.02$ μmol mol$^{-1}$ (Fig. 5 a and b). The assigned values of the calibration gases have currently a reproducibility of $0.01$ μmol mol$^{-1}$ (1-sigma). If we assume independent errors between the old C2 and the replacement C2, their difference can be expected to be within $0.01 \cdot \sqrt{2} = 0.014 \, \mu mol \, mol^{-1}$ at 1-sigma, which is smaller than the difference between the two periods. Because of that and since the noise grew as well, the change is most likely caused by the exchange of the analyzer. A change of the C3 at the end of September 2017 didn't have any significant influence on the precision or accuracy. However, despite the small bias, the accuracy and precision are still excellent for a NDIR $CO_2$ measurement system. Furthermore, since we are only interested in relative changes of $CO_2$ over the life time of a cylinder, the small bias is only of minor importance. The repeatability is much more meaningful because it shows the detection limit of our experiment.

### 3.2 Low flow measurements

In the low flow mode, 38 full tanks were depleted with vertically positioned cylinders. All measurements followed a similar pattern with a very small, almost linear $CO_2$ mole fraction increase down to about 30 bar that becomes much stronger from there. A fit following Langmuir's adsorption/desorption model was calculated for each measurement. The fit functions were used to calculate the average $CO_2$ enrichment with decreasing pressure using the pressure measurements, it is $0.089 \pm 0.013$ μmol mol$^{-1}$ (Fig.6), the given error corresponds to the standard error (1-sigma) of the average. Since each cylinder started with a different pressure, the averages could be not entirely comparable. However, if the enrichment is calculated over the same pressure span of 150 to 1 bar, the result is again $0.090 \pm 0.009$ μmol mol$^{-1}$, which is the same within the given uncertainty. As mentioned in the methods, the coefficient K of the Langmuir model had a fixed value of $0.001$ bar$^{-1}$, but also the value of the initially adsorbed $CO_2$, $CO_{2,ad}$ was relatively constant throughout all measurements, the average was $0.0165 \pm 0.0016$ μmol mol$^{-1}$ at the initial pressure. When fitting a function based on Rayleigh distillation, the average of the fractionation factor α is $0.999957 \pm 0.000004$, which would cause a $CO_2$ increase of about $0.085$ μmol mol$^{-1}$ if the pressure drops from 150 to 1 bar. The standard deviation of the average α is very low. If the measurement system's repeatability as deducted from the target gas measurements is taken into account, a realistic error of α should be about four times bigger.

Two additional low flow measurements with horizontally positioned cylinders were done. Again the Langmuir fit functions of the two measurements were used to estimate the average $CO_2$ enrichment, which was



0.019 ± 0.003 µmol mol$^{-1}$ for the measured pressure drop and 0.021 ± 0.004 µmol mol$^{-1}$ for a pressure drop from 150 to 1 bar, respectively, which is a hardly significant considering the detection limit of the measurement system. One of the two cylinders was equipped with thermistors, similar to the high flow setup, representing at the same time the only low flow measurement with temperature measurements (Fig. 4 b). The temperature measurements didn't reveal

any features related to the pressure drop in the cylinder. The observed periodical cylinder temperature changes with an amplitude of about 1 K were mainly driven by changes of the room temperature due to the air conditioning regulation and not by the gas decanting (Fig. 7).

**3.3 High flow measurements**

In the high flow mode, eight complete drainings were done with cylinders vertically positioned. The average

enrichment calculated from the Langmuir-only fits corrected to a pressure drop from 150 to 1 bar was 0.24 ± 0.04 µmol mol$^{-1}$ (Fig. 8). The average value for $CO_{2,ad}$ was 0.043 ± 0.008 µmol mol$^{-1}$, which is about 2.5 times bigger than the value found in the low flow experiments (K was again fixed at 0.001 bar$^{-1}$). If the combined Langmuir/Rayleigh fit is used with the Langmuir coefficients fixed from the low flow measurements, the enrichment adjusted to a pressure drop of 150 to 1 bar is 0.22 ± 0.05 µmol mol$^{-1}$, which is basically the same within

the uncertainties. The average fractionation factor α is 0.99993 ± 0.00002, which means $CO_2$ depletion in the outflowing gas. Subtracting the Langmuir function with the low flow coefficients from the measurements and using the residuals to plot $\ln(X/X_0)$ against $-\ln(P/P_0)$ yields an elongated cloud with slope of 1-α, indicating that Rayleigh distillation might be responsible for the additional $CO_2$ enrichment seen in the high flow measurements (Fig. 9). At the same time, the temperature development of the cylinder and the pressure regulator were measured (Fig. 10 a). At

the front end of the regulator, where the secondary pressure reduction happens, the temperature dropped rapidly by about 6.76 ± 0.59 K within 94 ± 22 min with the given errors being the standard error (1-sigma) of the average of all runs. The thermistors indicate also a slightly bigger temperature drop of 7.01 ± 0.62 K within 98 ± 23 min at the first stage compared to the second stage, although the difference is hardly significant. The stem of the regulator showed a reduced temperature drop of about 4.58 ± 0.44 K with the minimum delayed by about 133 ± 33 min. The cylinder

valve follows much closer the temperature of the cylinder body than the temperature of the regulator. It shows a drop of 2.33 ± 0.25 K with the minimum occurring 234 ± 25 min after the beginning of the gas flow. The temperature of the cylinder was measured at five evenly distributed levels. The average temperature drops from the top to the bottom level were 2.29 ± 0.23 K, 2.40 ± 0.19 K, 2.53 ± 0.17 K, 2.58 ± 0.18 K and 2.55 ± 0.21 K, respectively, with the minima occurring at 251 ± 29 min, 266 ± 23 min, 281 ± 17 min, 286 ± 16 min and

283 ± 22 min after the gas flow was initiated. The temperatures of the different levels of the cylinder body follow each other closely until one after the other reaches the minimum. Then they start to fan out until the end of the experiment, reaching a spread of 0.55 ± 0.13 K with the level near the ground showing the lowest temperature and the level near the shoulder showing the highest temperature.

Three more complete drainings were done with the cylinders horizontally positioned. In the beginning of the

experiments, the temperature of the regulator, the stem and the valve show a fast drop to minimal temperatures followed by a slow gradual temperature increase, similar as with the vertically positioned cylinders. Also the





cylinder valve follows closely the temperature of the cylinder body, although it seems to cool down a bit more and appears to be slightly more influenced by the regulator stem compared to the measurements with the vertical cylinders. The maximal temperature drops of the regulator from the front end to the stem are $6.40 \pm 0.18$ K, $6.72 \pm 0.22$ K and $4.66 \pm 0.31$ K, respectively with delay times of $112 \pm 16$ min, $109 \pm 21$ min and $194 \pm 43$ min

after starting the gas flow. The cylinder valve shows a temperature drop of $2.81 \pm 0.52$ K with a time delay of $251 \pm 15$ min. The temperatures of the cylinder body show a different behavior. If the corresponding thermistors of the lower and the upper side are averaged, similarly to the measurements with vertically positioned cylinders, the temperature drop seems to be the same everywhere within the uncertainties. From the valve end to the bottom end of the cylinder the average temperature drops of the three measurements are $2.82 \pm 0.64$ K, $2.80 \pm 0.59$ K,

$2.83 \pm 0.67$ K, $2.80 \pm 0.67$ K, and $2.78 \pm 0.67$ K, with time delays of $292 \pm 38$ min, $292 \pm 36$ min, $291 \pm 35$ min, $299 \pm 33$ min and $299 \pm 35$ min, respectively (Fig. 10 b). The given errors correspond to the standard deviation (1-sigma) of the average over the three runs. However, if the temperature development of each thermistor is evaluated individually, interesting details can be found. Since the three cylinders had slightly different starting pressures, the pressures had to be converted into a relative pressure measure in order to make them comparable. To

do so the pressure at each measurement was divided by the initial pressure ($P/P_0$) which means 1 (or 100 %) stands for a full cylinder, 0 means the cylinder is empty. The temperature measurements of each thermistor during the three runs were matched according to the relative pressure and to make the temperatures comparable, the difference of the temperature measured by the individual thermistor minus the cylinder body average (including the valve) was calculated. The results of the three runs were averaged and the standard deviation calculated. On the cylinder body,

the temperature of the upper thermistor is usually higher than the temperature of its counterpart at the lower side (Fig. 11). The temperatures of the two thermistors at the valve are the same within uncertainty. In the beginning at 100 % relative pressure, there is no significant temperature gradient along the cylinder body. At 75 % relative pressure, the front end is cooler than the cylinder body and the difference between the upper and the lower side starts to grow with the largest difference of 0.25 K being in the middle of the cylinder body. At 50 % relative pressure the

temperature distribution becomes symmetrical, with the lowest difference at the front end and the largest difference still in the middle of the cylinder body. There the temperature difference is about 0.3 K and remains stable for the rest of the experiment. This is also the point where the $CO_2$ depletion starts to follow the 1-$\alpha$ slope in the logarithmic plot indicating Rayleigh fractionation as will be shown later in this section (Fig. 9). At 25 % relative pressure the valve and the shoulder start to warm up (Fig. 11). While the temperature gradient along the cylinder becomes

smaller on the upper side, the gradient along the lower side is increasing. The largest difference is again at the middle of the body, it is still about 0.30 K. At 1 % relative pressure, the temperature gradient at the upper side almost vanished while it is the largest now for the lower side. The largest temperature difference of about 0.30 K is still at the middle of the cylinder body. The mole fraction measurements of these cylinders looked completely different. From the start of the measurement down to about 30 bar, the $CO_2$ mole fraction of all three cylinders

showed first a slight decrease of about 0.05 µmol mol$^{-1}$, followed by a small increase back to the original $CO_2$ level. From 30 bar until the end of the measurement, the $CO_2$ measurements show a steep $CO_2$ depletion (Fig. 12). If only the Langmuir function was used, $CO_{2,ad}$ had to become negative, which is physically impossible. Using the





combined Langmuir/Rayleigh fit function with the Langmuir coefficients fixed from the low flow setting gives us an average $CO_2$ depletion of $0.20 \pm 0.03$ µmol mol$^{-1}$ over a pressure drop from 150 to 1 bar. The average fractionation factor α is $1.00014 \pm 0.00003$ indicating a $CO_2$ enrichment in the outflowing sample gas, in contrast to the vertical cylinders (Fig. 9). The slope still follows 1-α, consistent with Rayleigh fractionation. In all three measurements the

logarithmic plots show a flat plateau in the beginning. The decrease starts in all three measurements at -ln(P/P$_0$)≈0.7, which corresponds roughly to a half empty cylinder. If only the $CO_2$ measurements below 50 % of the cylinder's pressure are used to calculate α, then the average fractionation factor for the outflow becomes $1.00021 \pm 0.00004$, indicating an even stronger fractionation. When the Rayleigh fractionation with the stronger fractionation factor is only applied after the cylinder is half empty, when the temperature difference between the upper and the lower side

of the cylinder body reaches its final value of 0.3 K, the average final depletion is $-0.26 \pm 0.07$ µmol mol$^{-1}$.

### 3.4 Moving cylinders into different orientations while measuring

One high flow run each was done with a cylinder being in horizontal position and then put into vertical position at about 30 bar and vice versa. The tank that was first in horizontal position showed a very stable $CO_2$ mole fraction ($411.00 \pm 0.02$ µmol mol$^{-1}$) down to 32.7 bar when it was put up into a vertical position (Fig. 13). With the

movement, the $CO_2$ mole fraction dropped within 15 min by about 0.08 µmol mol$^{-1}$ and from there on, it showed similar enrichment behavior as the other measurements of vertically positioned cylinders with a $CO_2$ increase of about 0.12 µmol mol$^{-1}$. Also the opposite experiment showed stable $CO_2$ mole fractions ($401.90 \pm 0.02$ µmol mol$^{-1}$) until it was laid down at 32.1 bar. As soon as it was in a horizontal position the $CO_2$ mole fraction jumped up within 25 min by about 0.13 µmol mol$^{-1}$ and decreased in a similar manner as the other horizontally positioned cylinders

with a $CO_2$ decrease of about 0.14 µmol mol$^{-1}$. Interestingly, the two measurements seem to mirror each other pretty well (Fig. 13). The temperature development of the regulators looks the same as for other high flow measurements and the temperature of the cylinder body shows similar main characteristics, such as a fast drop in the beginning and a slow increase after reaching a minimum. However, there are some interesting differences. Until the cylinders are moved, the temperature measurements follow the usual individual pattern, the temperatures of the vertical cylinder

drop and fan out, the ones of the horizontal cylinder drop but stay together. After the cylinders are moved, the temperature measurements of the cylinder that is now horizontal converge (Fig. 14 a) while the ones of the cylinder that is now vertical fan out (Fig. 14 b).

Three additional runs were done, where the cylinders were first in horizontal position, and then put in a vertical position but with their valves at the bottom. During the first run, the handling of the cylinder didn't go as smooth as

planned and the turning of the cylinder took several attempts. Also the data logger for the thermistors stopped after about two hours. The $CO_2$ mole fraction of the first run is stable at $401.74 \pm 0.02$ µmol mol$^{-1}$, after the manipulation at 30.7 bar it stepped up by 0.05 µmol mol$^{-1}$ and remained stable at $401.79 \pm 0.02$ µmol mol$^{-1}$ until the cylinder was empty. The second cylinder was also stable at $401.85 \pm 0.02$ µmol mol$^{-1}$ until it was put up on its valve side at 32.2 bar, then the $CO_2$ mole fraction dropped by 0.08 µmol mol$^{-1}$ to $401.77 \pm 0.02$ µmol mol$^{-1}$ were it remained

stable until the cylinder was empty. The third cylinder again showed stable $CO_2$ mole fractions in the beginning it was at $401.83 \pm 0.02$ µmol mol$^{-1}$. At 27.8 bar it was put up and the $CO_2$ mole fraction dropped by 0.10 µmol mol$^{-1}$





were it stayed at $401.72 \pm 0.02$ µmol mol$^{-1}$ until the cylinder was empty. The temperature measurements of the two cylinders show the same behavior and they are comparable to the ones where the horizontal cylinder was brought into a vertical position. The only difference is that after the fanning out of the different temperature levels, the temperatures at the bottom of the cylinder are the highest, the temperatures at the valve, which is here the lower end, are the lowest (Fig. 14 c).

**3.5 Heating cylinders**

Two horizontally positioned cylinders were measured with constant heating starting at a cylinder pressure of 30 bar. The first cylinder showed a stable $CO_2$ mole fraction before heating, it was $410.36 \pm 0.03$ µmol mol$^{-1}$. The resumed $CO_2$ measurement after the cylinder temperature reached 30 °C showed a drop of about 0.09 µmol mol$^{-1}$ and remained stable at $410.27 \pm 0.02$ µmol mol$^{-1}$ until the cylinder was empty (Fig. 15). The second cylinder showed no changes in the $CO_2$ mole fraction before and during the heating. The average $CO_2$ mole fraction down to 30 bar was $410.99 \pm 0.03$ µmol mol$^{-1}$, after the temperature reached the preset value it was at $410.99 \pm 0.02$ µmol mol$^{-1}$. The temperature measurements of the two cylinders are the virtually the same (Fig. 16 a). In the beginning, they show the same pattern as the other high flow measurements with horizontally positioned cylinders, a temperature drop with the onset of the gas flow and almost no dispersion of the temperatures along the cylinder. At 30 bar, the heating began and the temperature increased and overshot slightly. Since the thermostat was attached to the shoulder of the cylinder, the temperature measured there is closest to the preset 30 °C. From there it increased by about 4.5 K with a maximum at the bottom of the cylinder. The cylinder wall is slightly thicker at the shoulder resulting in a bigger thermal mass. That, and the fact that there was a heat band at the bottom, might be why the temperature is higher at the bottom than at the shoulder and why the temperature overshot in the beginning of the heating.

Two vertically positioned cylinders were measured with burst heating up to 30 °C, starting at 50 bar. The first cylinder didn't seem to be affected by the heat burst, before heating the $CO_2$ mole fraction was stable at $401.91 \pm 0.02$ µmol mol$^{-1}$, after heating the $CO_2$ mole fraction followed the same pattern as with vertically positioned high flow experiments with no heating, resulting in a $CO_2$ enrichment of 0.15 µmol mol$^{-1}$. Unfortunately, the measurement cycle started a full calibration at about 52 bar, which is why there are no $CO_2$ data while heating. Also the temperature data logging stopped working after four hours, missing the interesting part of the experiment. In the second measurement, the $CO_2$ mole fraction was stable at $401.71 \pm 0.02$ µmol mol$^{-1}$ before the heating was switched on. With the beginning of the heating, the $CO_2$ mole fraction increased by about 0.10 µmol mol$^{-1}$, but again a full calibration obscures partly what happened during the heat burst. When the heaters were turned off, the $CO_2$ mole fraction fell back on values similar to high flow runs with vertical cylinders without heating and followed their enrichment pattern from there. The enrichment from the beginning of the heating until the cylinder was empty corresponded to 0.13 µmol mol$^{-1}$ (Fig. 17). Initially the temperature development looks about the same as for other vertically positioned cylinders, the temperature is coolest at the bottom and highest at the shoulder. Then, after the heating was switched on, the temperature gradient was turned upside down almost immediately. The set temperature of 30 °C was reached after about one hour which corresponds to a pressure decrease of about 10 bar. As soon as the





## 4. Discussion

The low flow measurements with cylinders vertically positioned show repeatedly comparable $CO_2$ enrichment with
decreasing pressure, no matter which cylinder was measured. That suggests that the observed $CO_2$ enrichment may
be universal for this type of aluminum cylinder. The only low flow temperature measurement available was done on
one of two vertically positioned cylinders. Towards the end of these measurements, the laboratory's air conditioning
wasn't very stable, varying by up to 1.5 K. The temperature variation is also visible in the $CO_2$ mole fraction of two
cylinders measured, making it impossible to calculate the $CO_2$ enrichment of this run properly. However, during the
first few days, when the background temperature was more stable, the temperature measurements also reveal that the
slow pressure drop of the low flow setting does not cause a big temperature drop in the cylinders. Therefore the $CO_2$
enrichment in the low flow experiments is most probably not temperature driven, but rather caused by $CO_2$
desorbing from the walls with decreasing pressure, following Langmuir's adsorption/desorption model. Assuming a
pressure of 150 bar, a K value of 0.001 bar$^{-1}$ and using these values in Langmuir's equation predicts occupation of
the available wall spaces of about 13 %. By using a very simplified geometrical approach, this number becomes
even smaller. Assuming the inner surface of the cylinder $A_{cyl}$ is 0.75 m$^2$, the area a $CO_2$ molecule occupies
corresponds to the collision diameter squared ($D_{CO2}$ =0.39 ·10$^{-9}$ m), the number of molecules per mole is defined as
6.022·10$^{23}$ mol$^{-1}$ (Avogadro's number), a pressure of 150 bar, a temperature of 293.15 K, and using the
$CO_{2,ad} = 0.0165$ µmol mol$^{-1}$ from the low flow measurements, the fraction of occupied spaces can be calculated to be
$\frac{CO_{2,ad} \cdot D_{CO2}{}^2 \cdot P \cdot V_{cyl} \cdot N_A}{A_{cyl} \cdot R \cdot T} = 37$ %. There is not enough information in the data to determine which of the two numbers is
closer to reality. As mentioned in the methods section, a range of solutions can reproduce the observed enrichment
at 1 bar. But since with higher K values the enrichment effect becomes more and more pronounced at lower
pressures, so that the observed shape can not be met, K and the corresponding coverage factor θ (at 150 bar) have to
be low. A second conclusion is that the aluminum cylinders are a good choice to store $CO_2$-in-dry-air mixtures. In
the case of gravimetrically prepared standards the $CO_2$ mole fraction is calculated by weighing the $CO_2$ and the air
that have been added to the cylinder. Because part of the $CO_2$ is adsorbed by the cylinder wall, the assigned $CO_2$
mole fraction of the sample gas might be overestimated, leading to a small bias in the calibration of $CO_2$
measurements if not corrected properly. This effect is likely worse with smaller cylinders, where the surface to
volume ratio is bigger.

The $CO_2$ enrichment in the high flow measurement was on average 2.5 times higher than in the low flow
measurements. This corresponds well with the $CO_{2,ad}$ value of 0.047 µmol mol$^{-1}$ found by Leuenberger et al. (2015)
in a similar experiment. However, since the cylinders for the low flow and the high flow experiments were prepared
the same way, there is no reason why the $CO_2$ adsorbed by the wall should be that much higher. Also the ratio of the
adsorption/desorption rate (K), although slightly temperature dependent, doesn't explain the difference of the $CO_2$
enrichment between the low and the high flow experiments. Following the van't Hoff equation





$K(T) = K(T_0) \cdot e^{\frac{E}{R}\left(\frac{1}{T} - \frac{1}{T_0}\right)}$ (van't Hoff, 1900), assuming a desorption energy of -10 kJ mol$^{-1}$ and using the maximum measured temperature drop of about 10 K, the coefficient K would only vary by about 10 %. But as mentioned in the methods section, the fit function is very insensitive to K anyway. A possible explanation for the stronger $CO_2$ enrichment might be thermally driven processes. As the air expands inside the cylinder because the pressure drops, it

will undergo adiabatic cooling. The cooling will be partially shared with the cylinder wall through circulation and diffusion of the air. The air will circulate because the air near the walls will tend to remain warmer than in the core. The temperature measurements during the high flow experiments with vertically positioned cylinders show a temperature drop of about 2.5 K at the cylinder surface caused by the pressure drop. The temperature difference between the different levels becomes gradually bigger and is about 0.5 K between the top and the bottom end at the

end of the measurement. This is consistent with cool air sinking in the cylinder while warmer air is rising. Assuming that there are only slow laminar flows in the cylinder and because air is a poor heat conductor, it is likely that the air inside the cylinder is forming a considerably cooler core. In equilibrium $CO_2$ will be depleted slightly in air that is warmer and in contact with air that is cooler by ~0.06 ppm/K at 400 ppm (Chapman and Cowling, 1970). The upright position of the cylinder might add to the effect by separating the warm and the cool end spatially. By

draining gas from the cylinder, the warm depleted air comes out first and leaves slightly $CO_2$ enriched air in the cylinder. The cooler $CO_2$ enriched air follows later. Plotting ln(X/X$_0$), corrected for Langmuir desorption using the low flow coefficients, against -ln(P/P$_0$), the points line up nicely with a slope of 1-α, supporting the idea of Rayleigh fractionation being partly responsible for the $CO_2$ enrichment in the high flow experiments (Fig. 9). We also have to remember that in this situation the cylinder air is not perfectly mixed any more, and the air leaving the cylinder does

not sample the cylinder uniformly.

Besides adsorption/desorption effects, Rayleigh fractionation seems to be at work in the high flow measurements with horizontal positioned cylinder as well, causing a net decrease in the $CO_2$ mole fraction with decreasing pressure. When looking at the logarithmic plot (Fig. 9), the points form first a plateau with stable $CO_2$ mole fractions. The points seem to indicate an onset of Rayleigh fractionation when the cylinder is half empty. This is also

the moment when the temperature difference between the lower and the upper side reaches its maximum, which is maintained until the end of the measurement. An approximate possible explanation might be found by consulting the temperature measurements (Fig. 11). The temperature gradients along the cylinder and between the upper and the lower side change with decreasing pressure. At the very start there is almost no gradient visible, neither along the cylinder nor between the upper and the lower side. Between the start and 75 % of the initial pressure, the

temperature measurements indicate a cooling at the cylinder's shoulder and at the lower side. The cooling of the shoulder is most probably induced by heat conduction from the cylinder through the cylinder valve to the even cooler regulator. When the cylinder is half empty, the whole temperature distribution starts to shift. The temperature gradient along the upper side seems to mirror the temperature gradient along the lower side with the difference in the middle of the cylinder body being highest. Until now, the cooler air was always close to the valve, while the warmer

air was at the upper bottom end of the cylinder. In the cooler air, $CO_2$ becomes enriched and is drained out first based on its proximity to the valve. Thereby the air remaining in the cylinder becomes slowly depleted in $CO_2$. Additionally, the warmer air at the upper bottom side might impair or even block off convection, enhancing the





depletion. As the pressure drop in the regulator becomes gradually less the cooling at the valve end becomes weaker and it starts to warm up, slightly affecting the shoulder, too. With the pressure decreasing further and the air at the valve end being removed steadily, the warmer air from the upper bottom of the cylinder that is now depleted in $CO_2$ gradually becomes sample air. A second factor might be that the colder air has slightly lower viscosity. At the end of

the measurement, the most depleted air from the farthest end of the cylinder is moved to the valve by expansion and causes the lower most $CO_2$ measurements. This observation will need to be explained by a model of the expansion and outflow, combined with circulation, heat conduction, and diffusive mixing in the cylinder.

When the cylinders are moved during the measurements, it becomes obvious that the air in the cylinder is separated into different air masses of different temperatures. If it is laid down from vertical, the vanishing gradient along the

cylinder and the emerging temperature difference between the now lower and upper side are proof of that the jump in the $CO_2$ mole fraction when a cylinder is laid down is in accordance with the aforementioned thermal diffusion fractionation that $CO_2$ gets enriched in cool air that accumulates at the bottom of the cylinder. By laying it down, the cool $CO_2$ enriched air flows along the cylinder to the valve and is drained, while the $CO_2$ depleted warmer air goes to the upper side of the cylinder. The cool air gets warmed by the cylinder wall and a weak convection is started that

mixes the lower layers of air in the cylinder. With further decreasing pressure and gas expansion the depleted air from the upper side gets mixed into the drained air, thereby causing the $CO_2$ decrease measured by the system. In the opposite case in which the cylinder is horizontal first and then put up into a vertical position, the cool air sinks to the bottom and the warm depleted air goes up to the top where it gets drained first. That causes the initial $CO_2$ drop after the repositioning. The cooler $CO_2$ enriched air at the bottom gets again warmed by the cylinder walls, inducing

a weak convection. Due to the convection and the gas expansion caused by the decreasing pressure, $CO_2$ enriched air gets increasingly mixed into the drained air causing the measured increase of the $CO_2$ mole fraction.

With the three cylinders that were moved upside down, the picture is not very clear. Because of a logger failure, there are no temperature measurements on the first cylinder, which is why it will not be discussed here. The second and the third cylinders put upside down show a drop in the $CO_2$ mole fraction after they have been moved. While the

second cylinder remains stable at the slightly lower $CO_2$ mole fraction, the $CO_2$ mole fraction of the last cylinder goes up first by about $0.07\,\mu mol\ mol^{-1}$ from 30 to 15 bar and falls back by roughly the same amount until the end of the experiment. The temperature measurements of the two cylinders look the same. The only small difference between the two is the pressure when they were turned, the second cylinder was turned at 32.2 bar, the third cylinder was turned at 27.8 bar. However, whether this caused the different behavior in the $CO_2$ measurements remains

unclear.

The results from the experiments with constant heating from 30 bar to 1 bar don't draw a distinct picture. While the first cylinder doesn't seem to be affected by the heating, the second shows a drop in the $CO_2$ mole fraction and remains stable until the end of the experiment. Since the temperature gradient between the bottom and the valve end is quite large while heating, the $CO_2$ drop could be caused by mixing of the air masses due to convection induced by

the heating. But why only one cylinder shows that feature, while the other has a stable $CO_2$ mole fraction throughout the whole experiment remains unclear. Burst heating has only a short-term effect. Since the heat burst is not able to penetrate deep into the air in the cylinder it affects only the outermost layers. The temperature measurements show





that the cylinder becomes warmer at the bottom than at the valve end, probably due to the ninth heat band at the bottom and the slight cooling of the regulator at the valve end. This might cause the outermost $CO_2$ enriched layers from the bottom to rise and generate the measured $CO_2$ peak when it reaches the valve. The effect is transient, finishing before the heating is finished. Shortly after the heating is stopped, the inversed temperature gradient

returns to its usual distribution, supporting the assumption that the heat burst didn't penetrate deeply into the cylinder gas. This is also backed by the measurements of the $CO_2$ mole fraction that besides the short spike show a similar $CO_2$ increase as with vertically positioned cylinder in high flow mode.

## 5. Conclusion

The tested aluminum cylinders behaved always the same within uncertainties, the individual cylinders didn't show

distinct unique features. This is also true for the SGS cylinders, indicating no benefit in using these tanks in $CO_2$ measurements at ambient level. To describe the $CO_2$ enrichment in low flow settings, the Langmuir adsorption/desorption model using averaged coefficients is sufficient to describe the $CO_2$ enrichment effects in aluminum cylinders. This opens the possibility to use a general correction function in case a calibration cylinder on a field station runs empty. However, we still recommend changing calibration cylinders before the pressure drops

below 30 bar in order to avoid the steepest part of the enrichment at the lowest pressures, and the corrections that add uncertainty to the measurements. At the same time the currently recommended threshold of 20 bar (WMO, 2014) is supported by measurements of this study. Using the low flow coefficients for the Langmuir model, a drop from 150 to 20 bar results in a $CO_2$ enrichment of about 0.036 µmol mol$^{-1}$, which is still well within the WMO compatibility goal between laboratories.

In high flow settings additional thermal diffusion effects and Rayleigh fractionation come into play that overrule the simultaneously ongoing Langmuir adsorption/desorption. Depending on the positioning of the cylinder, $CO_2$ can be increasing or decreasing with decreasing pressure. However, this might be only the case for systems with a steady high flow. If cylinders are decanted in quick bursts with enough time in between to allow them to equilibrate thermally, thermal fractionation shouldn't be able to develop and only Langmuir adsorption/desorption effects have

to be taken into account. Some of the observed effects remain unexplained because the measurements were inconsistent, or the behavior of air in the cylinder needs to be modelled explicitly. To answer these questions additional controlled experiments would be necessary. A further benefit could be gained by using a CRDS (cavity ring down spectroscopy) gas analyzer because it doesn't need to be calibrated as often as an NDIR analyzer and it could measure several gas species simultaneously.

*Competing interests:* The authors declare that they have no conflict of interest.

*Data availability:* https://doi.org/10.15138/G3263N



*Acknowledgements:* The authors like to thank Philip Handley, Jonathan Kofler, Tim Newberger, Jack A. Higgs and Thomas Legard for their technical support to build and maintain the $CO_2$ measurement system as well as Allen Jordan and Emrys Hall for their help and expertise to build and run the temperature measurement system. Michael F. Schibig is supported by an Early Postdoc Mobility fellowship from the Swiss National Science Foundation (SNSF).



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





**Table 1 List of the cylinders used in this study and numbers and types of experiments done with each of them.**

| | | Cylinder Nr. | | | | | | | | | Total Nr. of measurements per setup |
|---|---|---|---|---|---|---|---|---|---|---|---|
| | | CB11795 | CB11873 | CB11876[1] | CB11906 | CB11941 | CB11976 | CB12009 | CC505453[2] | CC505457[2] | |
| **Low flow** | vertical | 5 | 5 | 1 | 5 | 4 | 5 | 5 | 4 | 4 | 38 |
| | horizontal | 1 | | | | | | | | 1 | 2 |
| **High flow** | vertical | 1 | 1 | | 1 | 1 | 1 | 1 | 1 | 1 | 8 |
| | horizontal | | 1 | | | | | 1 | 1 | | 3 |
| | Vertical to horizontal | | 1 | | | | | | | | 1 |
| | Horizontal to vertical | | | | | | 1 | | | | 1 |
| | Horizontal to vertical upside down | | | | | 1 | 1 | | 1 | | 3 |
| | Horizontal with heating | | | | 1 | 1 | | | | | 2 |
| | Vertical with burst heating | | | | | | | 1 | | 1 | 2 |
| | **Total Nr. of measurements per cylinder** | 7 | 8 | 1 | 7 | 7 | 8 | 8 | 7 | 7 | 60 |

[1] *Because of scratched cylinder valve the cylinder was replaced after the first measurement*

[2] *SGS (Superior Gas Stability) cylinders*





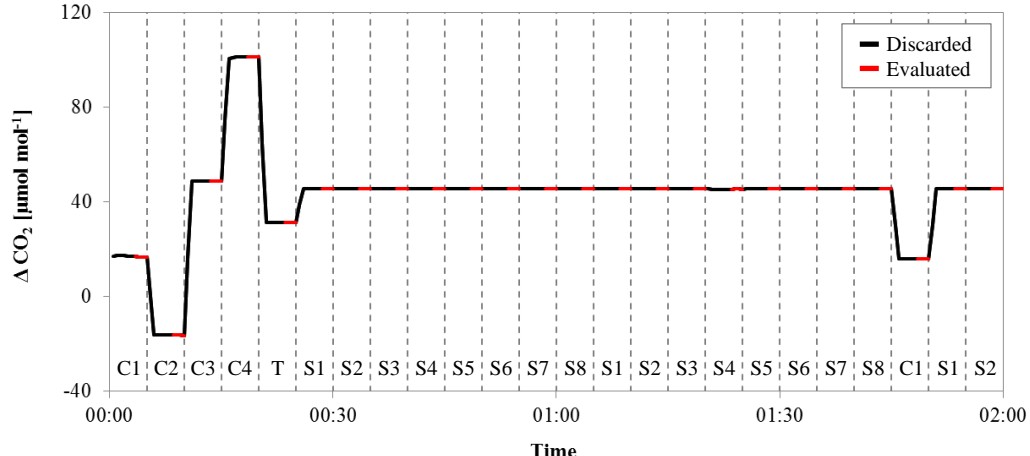

**Figure 1: Example of a calibration sequence from a low flow measurement. The x-axis represents time, the y-axis the delta signal of the NDIR analyzer. The gas measured is indicated by the codes at the top of the figure (C = Calibration, T = Target, S = sample), the switching of the valve is marked by the dashed vertical lines. Each gas was measured for 5 minutes, to avoid mixing and memory effects, the first 3 minutes were discarded (black lines) and only the last 2 minutes (red lines) were used for further evaluation.**



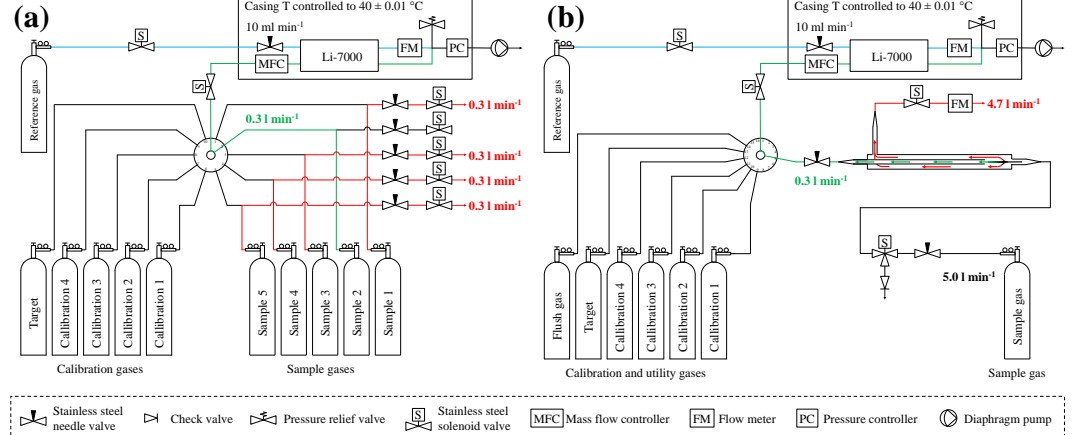

**Figure 2: Schematic of the low flow measurements system setup, here with the 10-port vici valve (a) and the high flow measurement system setup with the 16-port VICI valve (b). The red lines and numbers indicate the gas that is drained to the room to maintain a steady flow out of the cylinder, the green lines and numbers are what goes into the analyzer, the blue lines are the reference gas flow.**



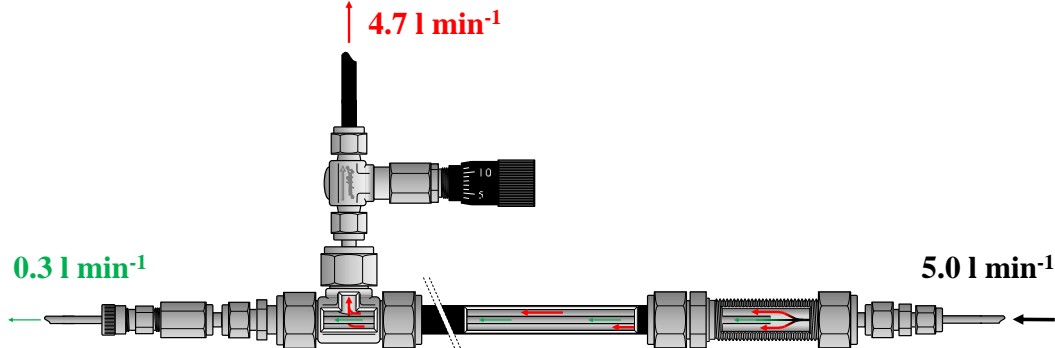

**Figure 3: Flow schematic of the high flow inlet system. The sample gas enters on the left side at 5.0 l min⁻¹. A small aliquot of 0.3 l min⁻¹ goes to the analyzer, the vast remainder of 4.7 l min⁻¹ goes to the exhaust. The ratio between the gas going to the analyzer and the exhaust, respectively, can be set by adjusting the needle valve on the exhaust side.**



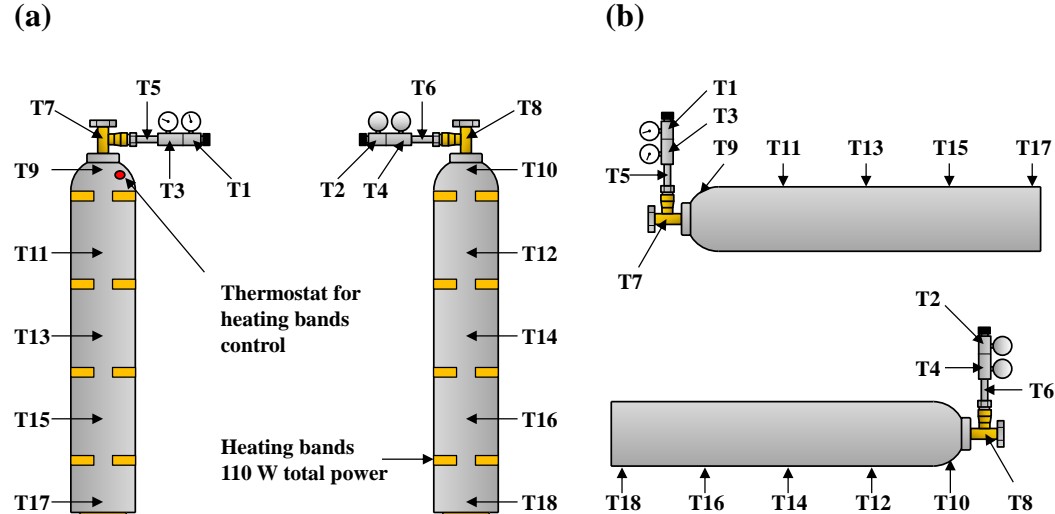

**Figure 4: Schematic of the locations of the thermistors on the cylinder in vertical (a) and horizontal positon (b) as well as the heating bands if used. Thermistors 19 to 25 were bundled and used to measure the background temperature of the laboratory.**



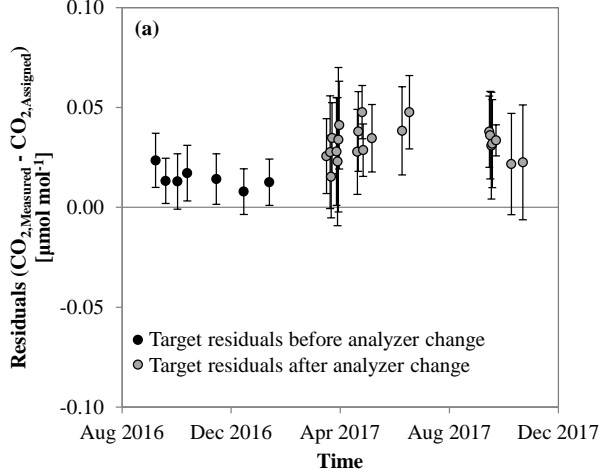

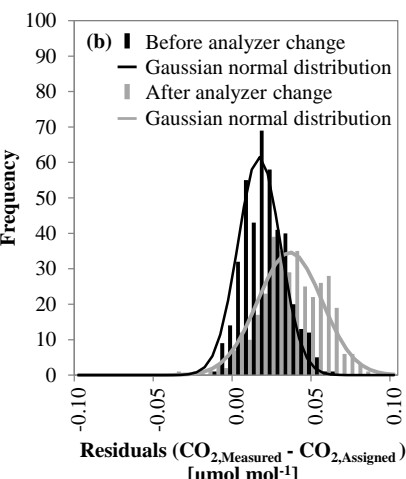

**Figure 5: Panel (a): Average of the difference of the target gas measurement minus the assigned CO$_2$ mole fraction for each run against time, black indicates the measurements before the analyzer change, grey after the analyzer change in March 2017. The error bars correspond to the standard deviation of the individual target gas measurements within each run. Panel (b): Histogram of the residuals of the target gas measurements, again, black stands for the measurements before, grey after the analyzer change.**





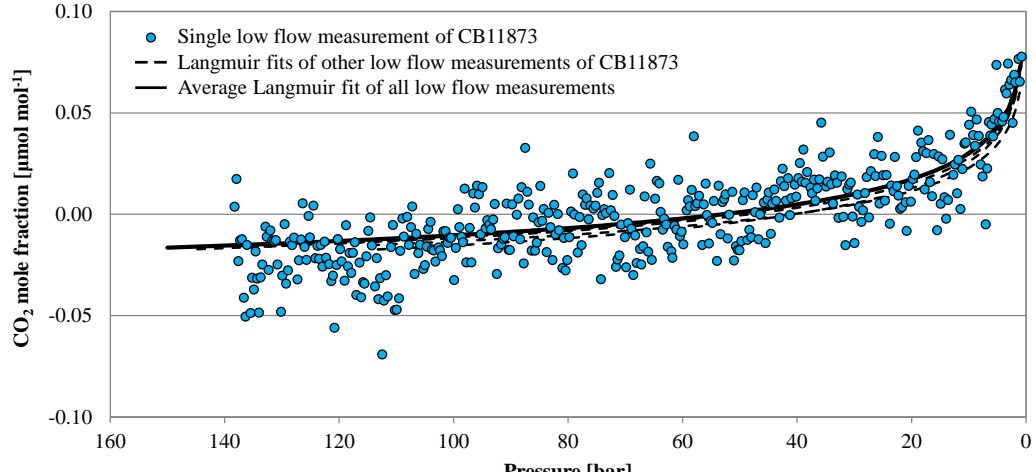

**Figure 6: The blue circles represent the CO₂ mole fraction measurement of a low flow experiment started on 17.10.2016 with CB11873 vertically positioned as a function of pressure, note the inverse pressure scale. The black dashed lines are the individual fits following the Langmuir model of the other low flow experiments done with CB11873 vertically positioned, the black solid line represents the average Langmuir fit using all low flow experiments with the cylinders vertically positioned. In order to plot all data in one plot, the corresponding CO₂,ini was subtracted from the measurements and the fits.**



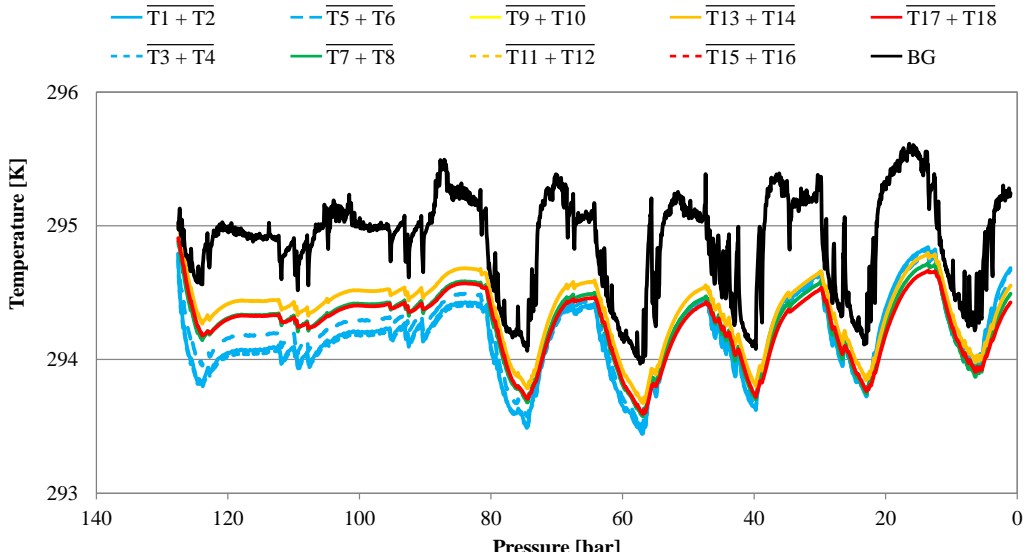

**Figure 7: Temperature development of CB11795 during the low flow measurement with vertically positioned cylinders as a function of pressure, note the inverse pressure scale. The lines represent averages of the following thermistors: Pressure regulator to stem: Solid blue: T1 and T2; dotted blue: T3 and T4; dashed blue: T5 and T6; Cylinder valve: green: T7 and T8; Cylinder body: yellow: T9 and T10; dashed orange: T11 and T12; solid orange: T13 and T14; dashed red: T15 and T16; solid red: T17 and T18; laboratory background: black: T19 to T25.**



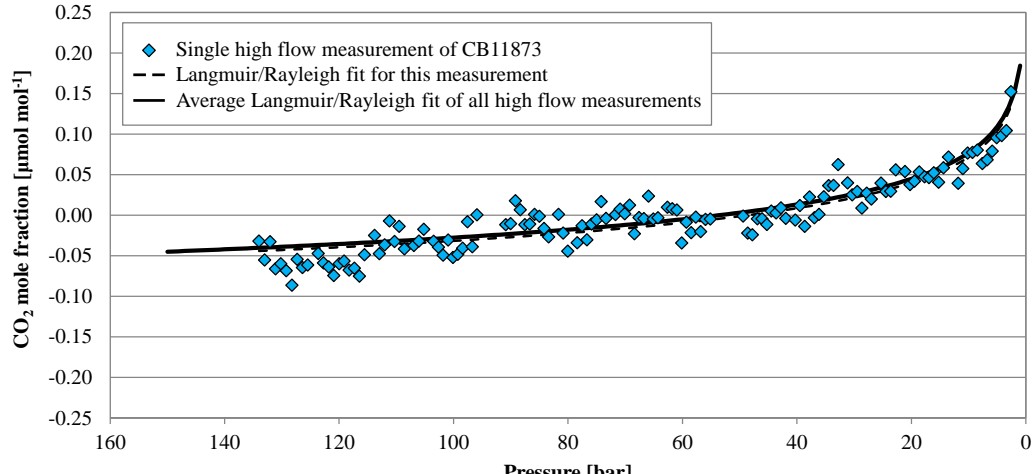

**Figure 8: The blue diamonds represent the $CO_2$ mole fraction measurement of the high flow experiment done on 5.4.2017 with CB11873 vertically positioned as a function of pressure, note the inverse pressure scale. The black dashed line is a fit following the combined Langmuir adsorption/desorption and Rayleigh distillation model, the black solid line represents the average of the combined Langmuir and Rayleigh fit using all high flow experiments with vertically positioned cylinders. In order to plot all data in one plot, the corresponding $CO_{2,ini}$ was subtracted from the measurements and the fits.**





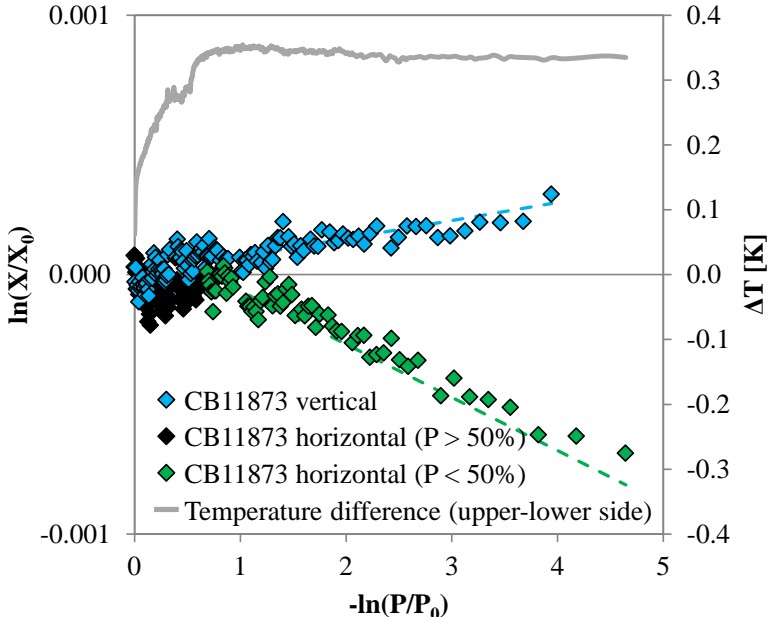

**Figure 9: CO$_2$ measurements of high flow experiments with CB11873 corrected for Langmuir adsorption/desorption effects based on the low flow experiments, vertically positioned cylinder in blue and horizontally positioned cylinder in black for the first half and green for the second half of the run. The data is plotted such that fractionations caused by Rayleigh distillation would follow a line with slope 1-α with α being the fractionation factor. These lines are indicated by the dashed lines in corresponding colors that were calculated based on an averaged α from all available experiments. The grey line plotted on the secondary y-axis is the temperature difference between T15 and T16 corresponding to the upper and the lower side of the horizontally positioned cylinder. It reaches its maximum after the cylinder is about half empty, which is when the fractionation seems to start.**





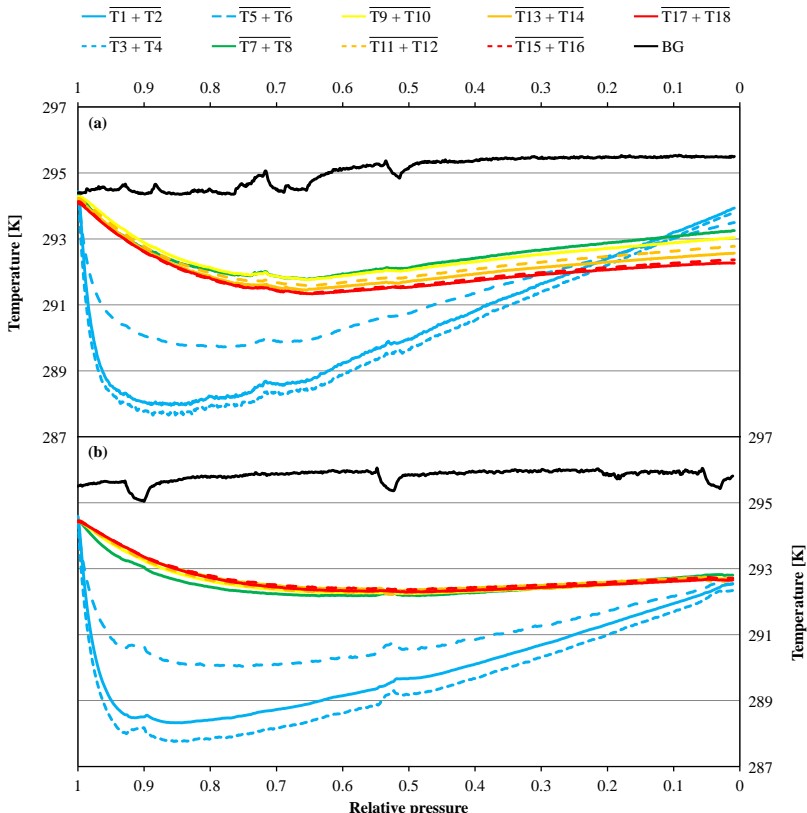

**Figure 10:** Temperature measurements against relative pressure, note the inverse x-axis. Because the initial pressures of the two cylinders were slightly different (CB11873 at 140.0 bar and CB12009 at 128.2 bar), the pressure is expressed as relative pressure ($P/P_0$) in order to use the same x-axes for both panels. The measurements in panel (a) come from a high flow experiment with vertically positioned cylinder (CB11873 on 4.5.2017), the temperatures in panel (b) were measured during a high flow experiment with horizontally positioned cylinder (CB12009 on 8.5.2017). The lines represent averages of the following thermistors: Pressure regulator to stem: Solid blue: T1 and T2; dotted blue: T3 and T4; dashed blue: T5 and T6; Cylinder valve: green: T7 and T8; Cylinder body: yellow: T9 and T10; dashed orange: T11 and T12; solid orange: T13 and T14; dashed red: T15 and T16; solid red: T17 and T18; laboratory background: black: T19 to T25.





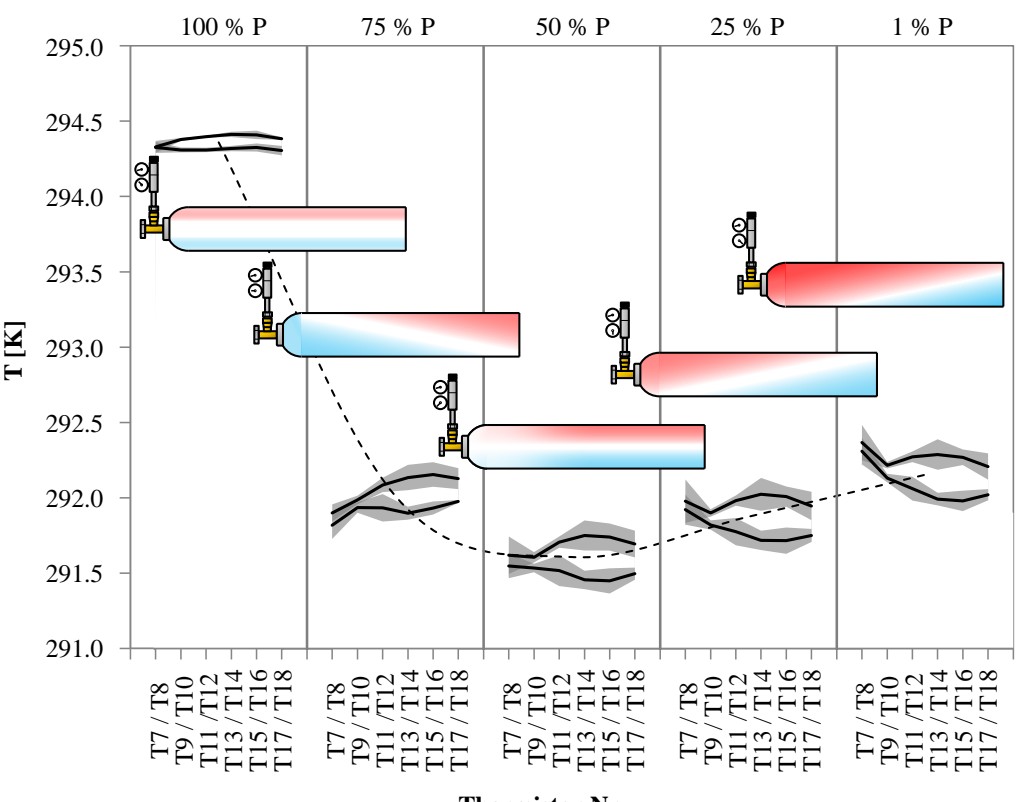

**Figure 11: Temperature difference between the upper and the lower side of the cylinder (in horizontal position) at five different stages of decanting. The temperature is given on the y-axis, the black lines are the average temperature along the upper and lower side of the cylinder derived from three runs, the shaded areas correspond to the standard deviation. The position of the temperature measurement along the cylinder is given for each stage individually on the x-axis in form of the thermistor number (see Fig. 4), the relative pressure is given as bins on the secondary x-axis. The black dashed line serves as an indicator for the general temperature development of the cylinder, it corresponds to the average of $T11$ to $T14$. The red and blue colors in the cylinders represent a possible distribution of warm (red) and cool (blue) air within the cylinder derived from the temperature measurements on the outside of the cylinder.**





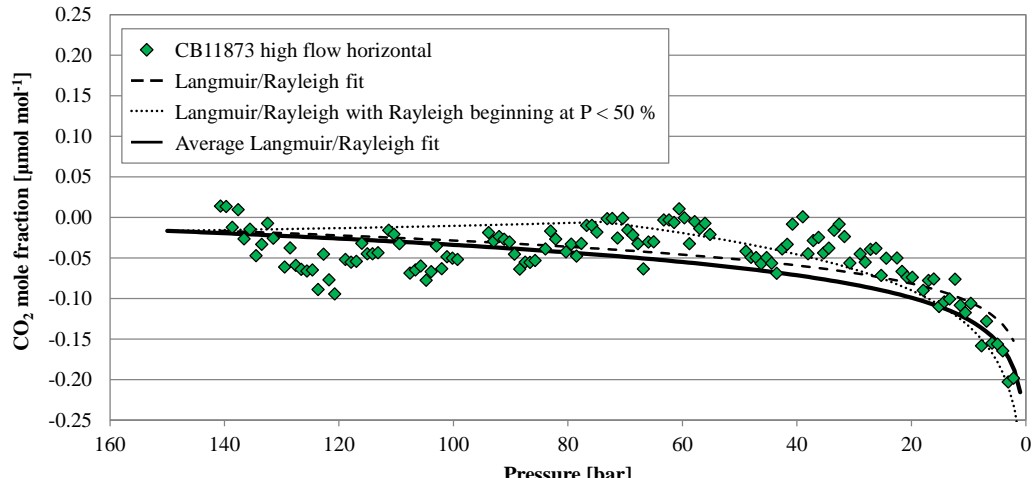

**Figure 12: The green diamonds represent the $CO_2$ mole fraction measurement of the high flow experiment done on 4.5.2017 with CB11873 horizontally positioned as a function of pressure, note the inverse pressure scale. The black dashed line is a fit following the combined Langmuir adsorption/desorption and Rayleigh distillation model, the black dotted line is a fit following the combined Langmuir adsorption/desorption and Rayleigh distillation model with the Rayleigh distillation starting when the cylinder is half empty, the black solid line represents the average of the combined Langmuir and Rayleigh fit using all high flow experiments with horizontally positioned cylinders. In order to plot all data in one plot, the corresponding $CO_{2,ini}$ was subtracted from the measurements and the fits.**





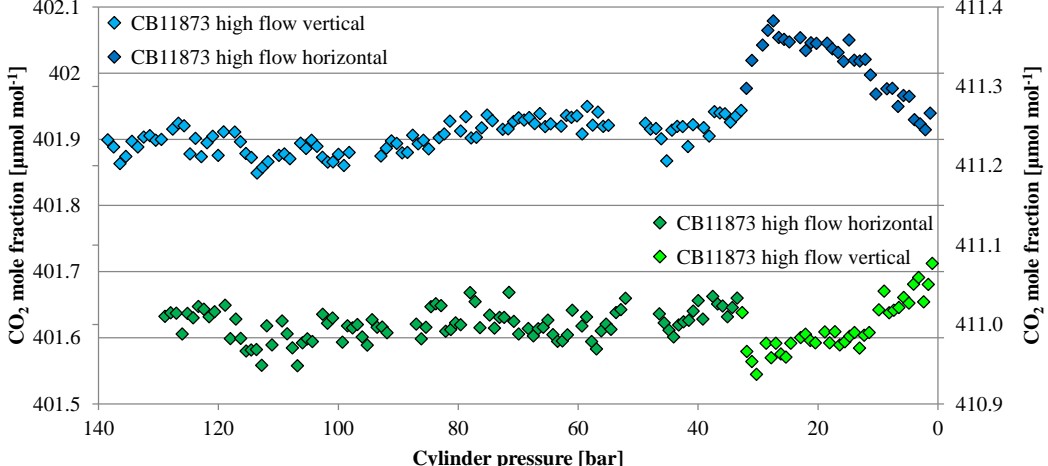

**Figure 13: CO$_2$ mole fraction as a function of pressure, note the inverse x-axis. The blue diamonds belong to the primary y-axis and were measured on CB11873 during a high flow experiment, where the cylinder was first vertically positioned and then laid down at 32.1 bar (indicated by the darker blue diamonds). The green diamonds belong to the secondary y-axis and were measured on CB11976 during a high flow experiment, where the cylinder was first horizontally positioned and then put up at 32.7 bar (indicated by the brighter green diamonds).**





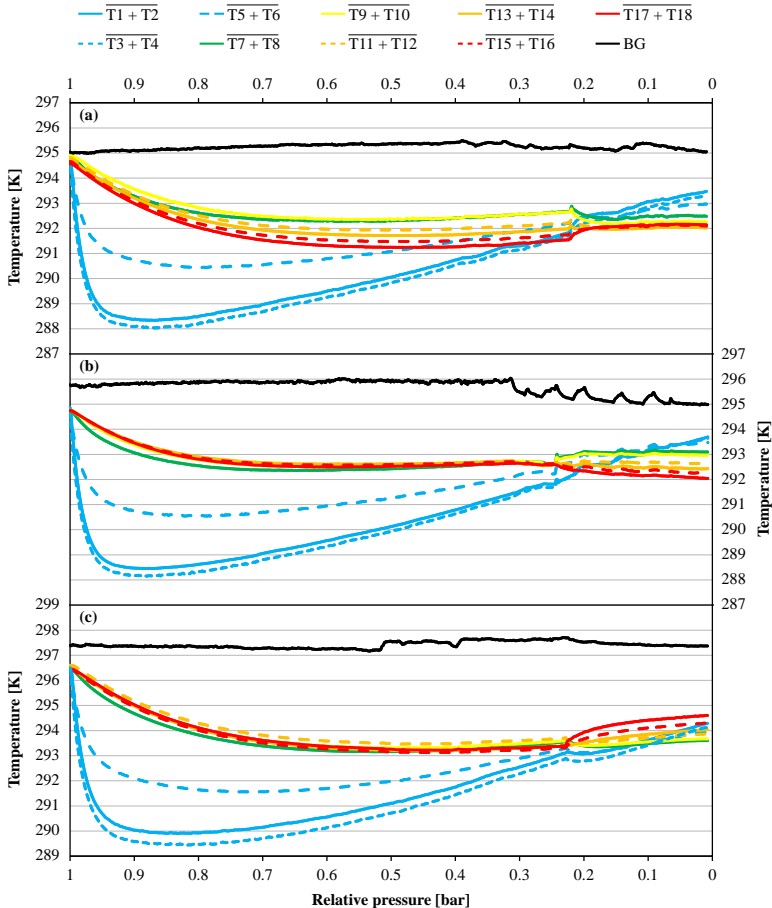

**Figure 14: Temperature measurements against relative pressure, note the inverse x-axis. Because the initial pressures of the cylinders were slightly different (CB11873 in panel (a) at 138.4 bar, CB11976 in panel (b) at 128.9 bar and CB11976 in panel (c) at 133.9 bar, respectively), the pressure is expressed as relative pressure ($P/P_0$) in order to use the same x-axes for all three panels. Panel (a) shows the temperature measurements of a high flow experiment where a vertically positioned cylinder was laid down (CB11873 at 32.1 bar), panel (b) shows a high flow experiment, where a horizontally positioned cylinder was put up (CB11976 at 32.7 bar) and panel (c) shows the temperature of a cylinder that was horizontally positioned and then put upside down (CB11976 at 32.2 bar). The lines represent averages of the following thermistors: Pressure regulator to stem: Solid blue: T1 and T2; dotted blue: T3 and T4; dashed blue: T5 and T6; Cylinder valve: green: T7 and T8; Cylinder body: yellow: T9 and T10; dashed orange: T11 and T12; solid orange: T13 and T14; dashed red: T15 and T16; solid red: T17 and T18; laboratory background: black: T19 to T25.**





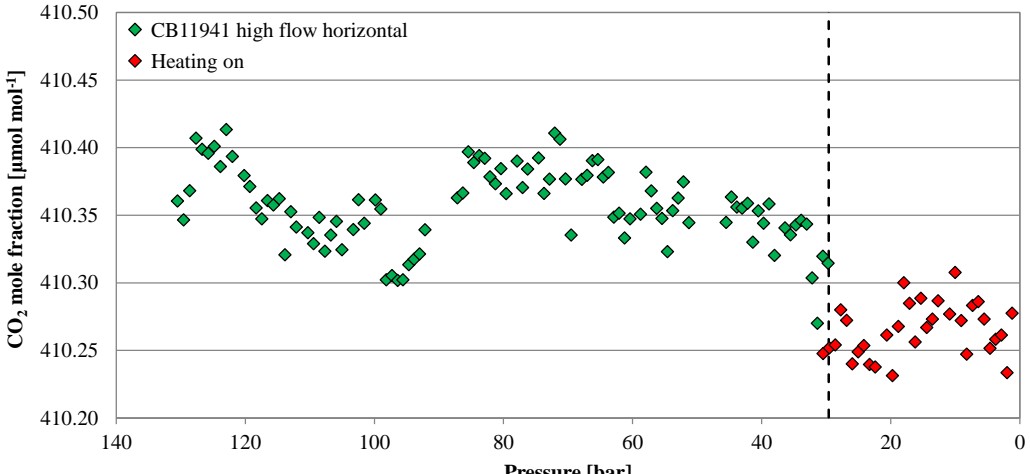

**Figure 15: CO$_2$ mole fraction of a high flow experiment as a function of pressure, note the inverse x-axis. The cylinder (CB11941 on 22.6.2017) was horizontally positioned throughout the whole experiment. The CO$_2$ mole fraction was measured down to 30 bar (green diamonds). At 30 bar (indicated by the dashed line), the flow was interrupted and the cylinder was heated up to 30°C, after the set temperature was reached, the CO$_2$ measurement continued (red diamonds). The heating caused a small pressure increase, which is why the first two points of the resumed measurements appear slightly above the 30 bar threshold.**





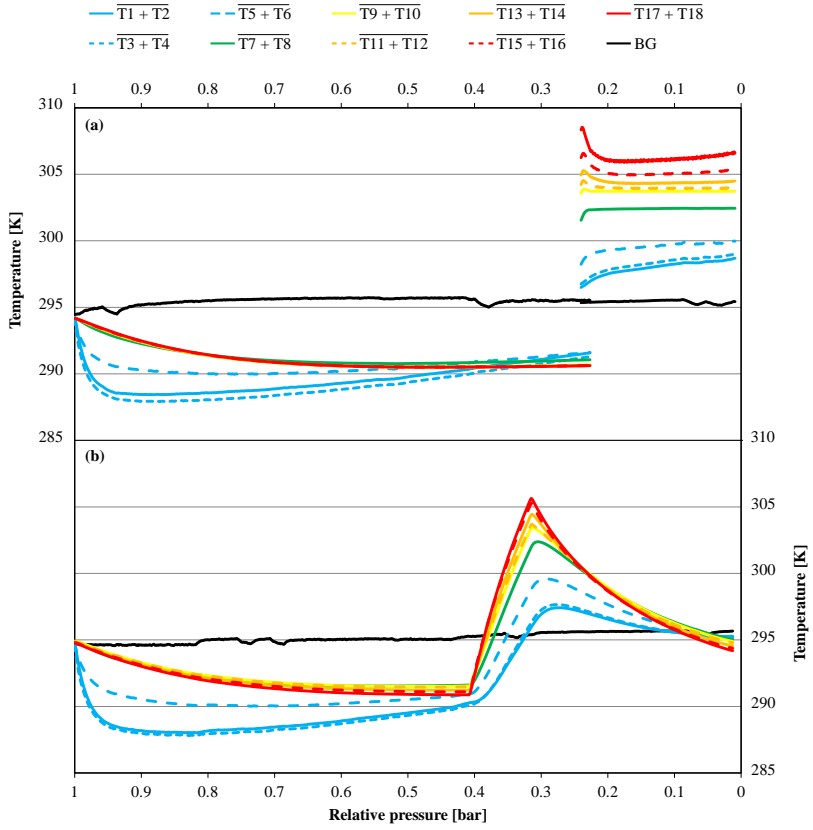

**Figure 16:** Temperature measurements against relative pressure, note the inverse x-axis. Because the initial pressures of the cylinders were slightly different (CB11941 at 130.5 bar and CB12009 at 124.0 bar, respectively), the pressure is expressed as relative pressure ($P/P_0$) in order to use the same x-axes for both panels. Panel (a) shows the temperature measurements of a high flow experiment where a vertically positioned cylinder (CB11941 on 21.6.2017) was measured until it reached 30 bar. At 30 bar, the flow was stopped and the heating turned on. After the thermostat read 30°C, the flow was switched back on and the measurement continued with the heating keeping it at a steady temperature until the end. The heating created a small pressure increase which is responsible for the small overlap in the x-axis, clearly visible in the background temperature. Panel (b) shows a high flow experiment, where a vertically positioned cylinder (CB12009 on 26.9.2017) got a burst of heat at 50 bar. The heating was maintained until the thermostat read 30°. After reaching the threshold, the heating was switched off (at 40.8 bar). During the burst heating, the $CO_2$ measurements continued. The lines represent averages of the following thermistors: Pressure regulator to stem: Solid blue: T1 and T2; dotted blue: T3 and T4; dashed blue: T5 and T6; Cylinder valve: green: T7 and T8; Cylinder body: yellow: T9 and T10; dashed orange: T11 and T12; solid orange: T13 and T14; dashed red: T15 and T16; solid red: T17 and T18; laboratory background: black: T19 to T25.





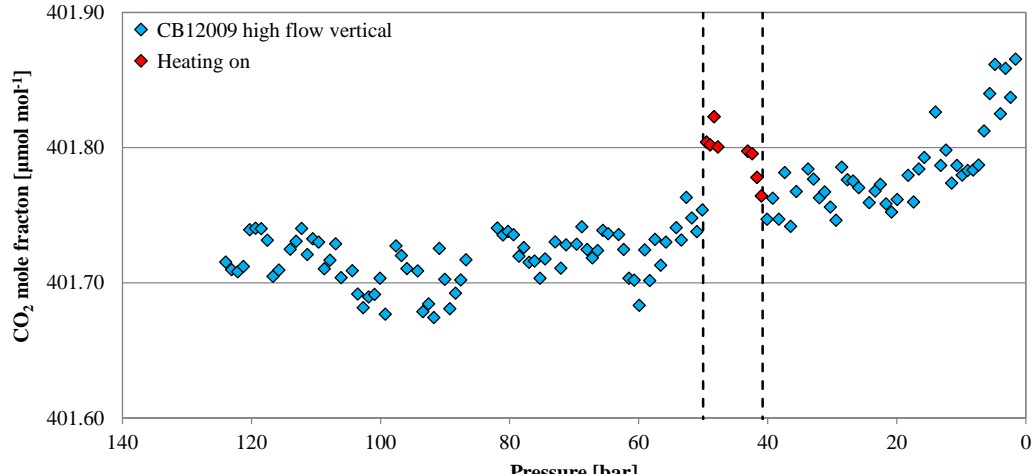

**Figure 17: $CO_2$ mole fraction of a high flow experiment as a function of pressure, note the inverse x-axis. The cylinder (CB12009 on 26.9.2017) was vertically positioned throughout the whole experiment. The $CO_2$ mole fraction was measured down to 50 bar (blue diamonds). At 50 bar (indicated by the first dashed line), the heating was switched on but the $CO_2$ measurements continued during the heating phase (red diamonds). After the heating's thermostat indicated that the cylinder reached 30°C the heating was switched off (indicated by the second dashed line) and the $CO_2$ mole fraction was measured until the end (again blue diamonds).**

