# Peer review of "Experiments with CO2 -in-air reference gases in high-pressure aluminum cylinders"

_Atmospheric Measurement Techniques, 2018_

## Referee Comment (RC1) · Anonymous Referee #1 · 20 Apr 2018

The manuscript "Experiments with CO2 -in-air reference gases in high-pressure aluminum cylinders" of M. Schibig et al. studies the stability of CO2 in air mixtures at ambient mole fractions. The topic is relevant, since accurate and reproducible measurements with traceability to standard scales are needed to detect changes in regional sources and sinks of CO2.

The manuscript is generally well written and concise. The chain of arguments is sound, and the topic is relevant for the scientific community. I therefore do recommend publication in AMT after addressing the following concerns.

General comments

The experiments carried out are clearly relevant for laboratory studies using large

amount of standard gases within a short time period. However, this is mostly not the case for long-term monitoring of CO2 and other research projects, where the gas of a standard cylinder is used over a much longer time period and in intervals. During period without use of standard gas, re-equilibration might take place, and the effect of changing CO2 mole fraction during the use of a standard that was observed in this study will in many cases not happen or be much less pronounced in a normal measurements set-up for ambient CO2.

Stability is much better for low flow conditions, but again, in reality, it might even be worse due to effects of the regulators. Especially in realistic measurement set-ups, this can be a problem, since only small aliquots of standard gases are used in longer time intervals, and the air is mostly sampled from the regulator and not directly from the cylinder. I recommend adding a few words on this issue.

The study was carried out with dry air mixtures (H2O < 1 $\mu$mol mol-1). Residual water content might have a significant impact on the behavior of CO2 absorption. Has the low water content be verified by measurements or other means? What will be the effect of residual water, even if less than 1 $\mu$mol mol-1? Could it be that differences in the residual water content explain at least partly the difference between individual cylinders or fillings?

A very recent publication studies similar effects including the influence of water on the stability of gaseous reference materials (Brewer et al., 2018). Citation of this work should be made in the final AMT version of the paper.

SGS (Superior Gas Stability) cylinders are mentioned in the introduction and methods, but no results are shown in the paper. In the conclusions, they are mentioned again, saying that they behave in the same way as untreated cylinders. This should also be shown and discussed in the results. E.g. individual fits could be shown for SGS and untreated cylinders in a separate figure similar to Fig.6.

I don't see much additional value of the experiments with heating and changing the
orientation of the cylinders during venting. The results of the heating experiments were not consistent between different runs, and probably more experiments would be needed to get a clear picture. For example, the mole fraction change after the start of the heating shown in Fig. 15 is not significantly different from changes observed at higher pressures during the same run. The results of the experiments with changing orientation during the draining are also based on only one run for each experiment, and it is unsure if they can be reproduced. The paper could be shortened and would improve if only the results of the low and high flow experiments, including different cylinder orientation and treatment, are presented.

I further recommend re-writing the conclusions. Currently, they are difficult to read without the full context of the paper, and present results which are not mentioned previously (e.g. SGS cylinders). Furthermore, the statement 'This opens the possibility to use a general correction function in case a calibration cylinder on a field station runs empty' should be made in the results section because it needs more careful discussion. Most likely, corrections will be associated with high uncertainties, since the calibration sequence at stations is different from your experiments in which the cylinders were emptied with a constant flow.

Specific comments

Page 2, line 13: kilogram is a SI unit, despite the fact that it is still based on an artifact. It should be removed from the list of examples in parenthesis.

Page 3, line 21ff: Add a short description of the performance of the analytical system (repeatability, drift etc.) here. This could be done by moving paragraph 3.1 to the method section.

Page 3, line 34: According to Fig. 1, C1 is repeatedly measured, not C2.

Page 9, first lines of result section and section 3.1 would better fit in the method section.

Page 10, section 3.2.: The low flow experiments are probably the most relevant for

most users of standard gases. Beside the average of all cylinders, the result of only one (representative) standard is shown, while in total 38 experiments of the same type were made. It would be valuable to see the variation between different cylinders / fillings, which could be added in either an additional figure or Fig. 6 (e.g. individual fits for all experiments). I recommend to also show and discuss the similar behavior of SGS and untreated cylinders could here.

Figures 6 and 8: The y-axis shows $\Delta CO_2$, and not $CO_2$ mole fraction, which needs to be corrected. Why do the measurements at higher pressures show a negative delta? Especially in Fig. 8 all $\Delta CO_2$ as well as the fits at higher pressures are negative. Is this correct?

Technical corrections

Page 2, line 24: the latest available GGMT report is not cited (WMO, 2016). It should be added. The format of the citations of the WMO reports needs also to be changed.

Page 2, line 28: replace 'SI values' with 'SI traceable values'.

Page 2, line 35: Cite the latest GGMT report here.

Page 3, line 36: Change to 'An additional full calibration was made at the end of each experiment'.

Page 6, line 25: Kitzis (2017) is missing in the references.

References

Brewer, P. J., Brown, R. J. C., Resner, K. V., Hill-Pearce, R. E., Worton, D. R., Allen, N. D. C., Blakley, K. C., Benucci, D., and Ellison, M. R.: Influence of Pressure on the Composition of Gaseous Reference Materials, Analytical Chemistry, 90, 3490-3495, 2018.

WMO: 18th WMO/IAEA Meeting on Carbon Dioxide, Other Greenhouse Gases and Related Tracers Measurement Techniques (GGMT-2015), La Jolla, CA, USA, 13-17

September 2015, GAW Report No. 229, World Meteorological Organization, Geneva, Switzerland, 2016.

---

## Referee Comment (RC2) · Anonymous Referee #2 · 7 May 2018

Atmos. Meas. Tech. Discuss., https://doi.org/10.5194/amt-2018-42

**Review of the submitted article:**

**Experiments with CO2 -in-air reference gases in high-pressure aluminum cylinders**

Michael F. Schibig, Duane Kitzis and Pieter P. Tans

General comments:

This paper describes a series of experiments performed to better understand previously observed drifts in the $CO_2$ mole fraction measured during the lifetime of reference gases in high pressure cylinders. The subject is of interest for worldwide measurements of atmospheric $CO_2$ mole fractions, which make use of such reference gases for their calibration. The authors have planned a consistent number of experiments to study the impact of various conditions. The measurements themselves appear robust, and all plots clearly summarise the observations. There is however some issue with the overall presentation, the organisation of the results, the link with the equations developed to fit the measurement results, and the relevance of some of the experiments. I therefore recommend a major revision before the paper can be published in AMT.

Although the general aim of the study appears to be a better understanding of already observed effects when using $CO_2$ in air standards, it is not so clear which particular questions are answered. For example, it is already recommended to leave not less than 20 bars in those cylinders, and this recommendation is again confirmed here. If that was the goal, it should be stated. The study also looked at the influence of the flow rate at which the gas is used, but the conclusions are never actually linked with some recommendation. Does this have any impact on their usage? Similarly, a big part of the paper is devoted to the analysis of thermal convections inside the cylinders. What is the impact here? Should this be followed by any recommendation, or was it only performed to better understand the process?

The organisation, terminology and wording of the paper can be improved. The description of the experiment is not very well structured and information on a same aspect is sometimes separated in different sections. At several occasions, words have been omitted, resulting in sentences which can still be understood but belong to oral language rather than a written paper. Authors also chose to sometimes use their own terminology, not following recommendations from international bodies such as IUPAC. It would be easier to follow with a more common terminology. It is also suggested to write all symbols for quantities in italic.

Finally, a recent paper dealing with the same subject was published earlier in 2018. It should be mentioned in the introduction and results compared in the discussion: Brewer, P. J., R. J. C. Brown, K. V. Resner, R. E. Hill-Pearce, D. R. Worton, N. D. C. Allen, K. C. Blakley, D. Benucci and M. R. Ellison (2018). "Influence of Pressure on the Composition of Gaseous Reference Materials." Analytical Chemistry 90(5): 3490-3495.

Specific comments by section:

1. Introduction: the end of that section needs to be revised to state more clearly which questions are being answered, which experiments were performed to do so, and which theory was applied. It is suggested to replace the assumptions with questions, such as i) is the effect cylinder−dependent? ii) in which conditions is the Langmuir model sufficient, and when does this need to be completed with the other effects already mentioned earlier in the section? iii) are SGS cylinders any better?

   In addition, other questions may be added if there are parts of the goal of the study, such as the influence of the flow rate at which the gas is sampled from cylinders, or the influence of the cylinder position.

   Finally, it is recommended to add a summary of the following sections, to clarify for the reader which information is provided in each section.

2. Section 2:

   a. The title ("methods") appears unclear. It actually contains information on the cylinders and analytical equipment, the measurement protocol, but also some theory to describe all processes at work.

b. It is suggested to modify the organisation, starting with a section only devoted to the description of the theory, then the equipment (cylinders plus gas lines plus instruments), and finally the measurement protocol. Within the protocol, the description of the measurement sequence can come first, followed by the explanation of the different conditions chosen to test the various assumptions.

c. The description of the protocol needs to be revised. Table 1 and figure 1 are clear and almost describe it completely, but the text brings more confusion. Specific words need to be chosen for each series of measurement (block, cycle, run), defined and further used with always the same meaning.

d. First part of the section 2 describes the measurement sequence, also displayed in Figure 1. Sequences were apparently setup to obtain both calibrated and a drift corrected values. This could be stated first, before providing the details of how this goal was achieved.

3. Section 2.4: is it relevant to show the equation used to determine the temperature? This cannot be checked and the only interesting value in that case seems to be the temperature.

4. Section 2.6: it is indicated that the filling was done at Niwot Ridge Station. Was this the case every time a cylinder was emptied? Was it sent back and refilled there? Please clarify.

5. Section 2.7: the entire section is difficult to follow, partly due to some considerations on unsuccessful attempt to fit the data. Even if some choices were justified by the observations, it would be easier to read if the different models were presented independently. The section could be divided in three parts, each of them presenting the assumptions, which part of the information comes from previous work (seems to be all in model 1 for example), which is the equation used during the fits, which parameters were assumed and which were fitted.
The presentation of the maths could be better balanced: while some common knowledge is sometimes detailed (such as the definition of the Avogadro number), some more expert information is not fully described (such as the Rayleigh distillation function).

a. Model 1: Langmuir. Equation (2) comes from Leuenberger 2015 where it was derived from the Langmuir model. This should be stated clearly. In addition, the Langmuir model normally uses partial pressure, not total pressure. Some consideration on the choice of using total pressure and its impact should be provided.
The explanation on the difficult fitting may be moved to the results, to limit this section to the theory. The quantity K was fixed at the same value than in the Leuenberger paper. Is that a coincidence? This should be better explained, but again not in this section, which should only state that K was fixed to find $CO_{2,ad}$.

b. Model 2: in this approach, it is stated that the fraction of occupied sites does not depend on the total pressure. Note that the $CO_2$ mole fraction varies with the total pressure, as demonstrated by the paper. It is certainly negligible but this should be explained.
This section introduces the symbol Na for a number of moles. This symbol being commonly chosen for the Avogadro number, it is recommended to replace with $n_a$.
The derivation of Equation (9) from (8) is not straightforward, compared to equation (5) for example which is very straightforward. It is suggested to provide the steps in annex, limiting this section to the model description and the equation used later on to fit the data.
The formalism could also be improved, avoiding a sentence such as "total trace gas" to replace a quantity. Why not defining $n_{tot} = n_{ad} + n_{gas}$.
Finally, like for the first model, considerations on results of the fit should be kept for the results analysis.

c. Model 3: in the third part, a Rayleigh distillation function is introduced. Some more explanation would be needed here, describing in one sentence the process and its expected impact on the $CO_2$ mole fraction.
The formalism may also be improved here: equation (12) introduces a mole fraction with the symbol $X$, where equation (13) uses $CO_{2,\,meas}$. It is suggested to always use $x$ for mole fractions.

6. Section 3:

a. This section is quite long and it is not always easy to distinguish the conclusions that are derived from each particular experiment. It is suggested to revise the structure, splitting in sub−sections to clearly identify what is the tested assumption, what were the observations, and the preliminary conclusion.

7. Section 3.1: there are many details in this section which may not be relevant for the understanding of the measurements. If the change of the analyser was not an issue, this should not be highlighted so strongly. The term accuracy does not seem appropriate, as the analyser was regularly calibrated. In addition it is noted at the end of the section that only relative measurements were performed. Then why not starting with this statement, and explaining that the repeatability (expressed with the standard deviation) of the measurements was evaluated, as this is the quantity which matters.

8. Section 3.2: it is mentioned here that the values $CO_{2,ini}$ and $CO_{2,ad}$ had a fix value. This was not so clear in the section on methods. It was also not clear how their values were actually chosen.

9. Section 3.3: it is suggested to separate this section in two, to treat vertical and horizontal cylinders separately. Also the reference to figure 9 appears too early, as this figure contains results with both positions of the cylinders. The clear distinction between them helps the understanding of the analysis, but maybe this could come after. Some effort to reduce the text, summarising each observation and drawing conclusions by step would be very much appreciated as well.

10. Section 3.4: please indicate which assumption was tested here and also when the cylinders were put upside−down. Certainly the idea of moving cylinders came from the observations during previous measurements, but this is not very clear.

11. Section 3.5: Some effort to reduce the text, summarising each observation and drawing conclusions by step would be very much appreciated as well here.

12. Section 4: it is suggested to also reshape this section, starting from each conclusion drawn during section 3, and summarising. At the end, some consideration on the impact should be given. For example, a possible small bias in the calibration of $CO_2$ measurement is mentioned. What is the conclusion? Will it be followed by further recommendation?
    At the end of page 15, it seems that some inconsistency is noted between the observations made when sampling the gas with low and high flow rates, as they would lead to two different values of the amount of $CO_2$ adsorbed on the wall. This appears quite serious. What is the conclusion? Does this question the entire model?

13. Section 5: the first conclusion from the low flow experiment appears to diminish the importance of the study. Indeed, if it is already recommended to stop using calibration gases when the pressure is below 20 bar, what was the purpose of this study?
    The second part would need to be strengthened; bringing more sounds conclusions and consideration on the implication (if any) on the usage of calibration gases.

Line-by-line comments:

**Page 1**

Line 20**:** "…In this study we found that during low flow conditions".  As this is the first mention of low flow rates, it should be stated more clearly that this is the flow rate used to sample the gas inside the standard.

Line 21: "showed similar $CO_2$ enrichment of $0.090 \pm 0.009$ µmol mol$^{-1}$ as the cylinder was emptied from about 140 to 1 bar above atmosphere". This is misleading, because the increase happens only after a certain pressure, not gradually during the cylinder was emptied. This is quite important as users could be afraid of using the standards.

Line 28: "In case they are used in high flow experiments that involve significant cylinder temperature changes, special attention has to be paid to possible fractionation effects". This is the only statement in the entire paper about possible impacts of sampling gas from the standards at a high flow rate. Some more consideration should be provided, both in the introduction to explain the current situation, and in the conclusion to provide a recommendation.

**Page 2**

Line 14-17**:** the sentence about the traceability explains the situation within WMO/GAW. It may not be extended to all $CO_2$ measurements in the world.

Line 18:"The resulting…" this sentence describes a goal rather than an observation. In addition the word "true" should be avoided and could be replaced with "unbiased".

Line 28:" SI values". Incorrect terminology. May be replaced with "values traceable to the SI".

Line 32:"accuracy". Incorrect terminology. To be replaced with Data Quality Objective or compatibility.

**Page 3**

Line 17**:** "To check the third hypothesis, two SGS cylinders were added to the set…" this would mean that 8 + 2 = 10 cylinders were tested. Table 1 shows only 8 cylinders (not considering the one which was replaced).

Line 30 and 32: the term "block" seems to indicate 10 measurements recorded during 5 minutes. Later in the text, the same quantity is sometimes only referred to as "a measurement". It can be understood to define a measurement as a 5 minute average, but this should be stated and always used in the same way.

**Page 4**

Line 2:" continuously at 10 ml min$^{-1}$ flowing reference cell signals". Revise the order of words in the sentence.

Line 16: C2 was measured twice? Should this be C2, C3?

Line 17: define what is "one run"

Line 19: "secondary". This term may be replaced here as it was already used previously in the traceability hierarchy.

Line 26:"before a block of C2 was measured". As the word "block" is not so appropriate to a 5 minute average, the entire sentence is difficult to understand.

Line 31:"lasts" should be "lasted".

**Page 5**

Line 28: "showed that they perform even slightly better." Sounds more like an opinion than an observation. Values should be provided, or the statement can simply be removed as it does not bring additional value.

**Page 6**

Line 15:"as soon as the cylinder reached 30 bar". Omission of the word "pressure", to be included.

**Page 7**

Equation (2): using the symbol of the molecule for its mole fraction is confusing when it is used in the text. It is recommended to follow IUPAC terminology.

Line 23:"amount density". This should be "amount concentration".

Line 24: "function of $CO_2$ only". This illustrates the problem of using the molecule's name instead of a symbol for its mole fraction.

**Page 8**

Line 2:"available". The term can be misleading, as it could indicate non occupied sites. May be replaced with "total number of sites".

Line 2:"amount of sites, expressed in moles". The mole is limited to an amount of molecules, atoms, ions, electrons, or other particles. This may be replaced with "maximum amount of adsorbed molecules", which also corresponds to the total number of sites.

**Page 10**

Line 23 to 25: revise the sentence, maybe using two distinct sentences.

Line 24:"standard error". A better term would be "standard deviation", as used later on in the text.

**Page 11**

Line 15: revise the last part of the sentence. It seems that some words are missing.

**Page 12**

Line 11:"errors". To be replaced with "uncertainties".

**Page 14**

Line 13: "are the virtually". Remove "the" and consider replacing the word "virtually", which seems more appropriate for a talk than for a paper.

**Page 15**

Line 15: "By using a ….the number becomes even smaller". Consider revising the sentence, first because the fraction of occupied sites goes from 13% to 37%, secondly because the second calculation seems to be an alternative one, so that two models are compared.

Line 17: "the number of molecules…Avogadro number". Is it really needed to state what is the Avogadro number?

Line 24:"a second conclusion": what was the first conclusion?

**Page 17**

Lines 9 to 12: consider splitting the sentence. This is currently difficult to understand.

**Page 18**

Line 12 and 13: "to describe the $CO_2$ enrichment". Consider removing one of the appearances of this part of the sentence.

Figures 6, 8, 12: the y−axis is a difference, not the $CO_2$ mole fraction. From the legend it seems that the first value was subtracted from the set, but then why is the first point not at zero?

---

## Author Comment (AC1) · 12 Jul 2018

The authors would like to thank Referee#1 and Referee# for their valuable comments and suggestions on this manuscript. Below we have addressed the individual remarks by each reviewer. The Referee's comments and questions are blue, the authors' replies are formatted as plain text, and excerpts from the manuscript as well as changes to the manuscript are given in italics.

**Reply to Anonymous Referee #1**

The manuscript "Experiments with  $CO_2$  -in-air reference gases in high-pressure aluminium cylinders" of M. Schibig et al. studies the stability of  $CO_2$  in air mixtures at ambient mole fractions. The topic is relevant, since accurate and reproducible measurements with traceability to standard scales are needed to detect changes in regional sources and sinks of  $CO_2$ .

The manuscript is generally well written and concise. The chain of arguments is sound, and the topic is relevant for the scientific community. I therefore do recommend publication in AMT after addressing the following concerns.

**General comments**

The experiments carried out are clearly relevant for laboratory studies using large amount of standard gases within a short time period. However, this is mostly not the case for long-term monitoring of  $CO_2$  and other research projects, where the gas of a standard cylinder is used over a much longer time period and in intervals. During period without use of standard gas, re-equilibration might take place, and the effect of changing  $CO_2$  mole fraction during the use of a standard that was observed in this study will in many cases not happen or be much less pronounced in a normal measurements set-up for ambient  $CO_2$ .

We disagree with the referee. The adsorption/desorption effect is mainly pressure driven and not dependent on time. According to Langmuir's adsorption/desorption equation, the equilibrium between the molecules on the cylinder wall and in the air has to change with changing pressure.

Stability is much better for low flow conditions, but again, in reality, it might even be worse due to effects of the regulators. Especially in realistic measurement set-ups, this can be a problem, since only small aliquots of standard gases are used in longer time intervals, and the air is mostly sampled from the regulator and not directly from the cylinder. I recommend adding a few words on this issue.

In realistic measurement set-ups at an e.g. atmospheric measurement station, the main valve of a cylinder remains open and therefore the gas in the stem and the regulator can equilibrate with the gas in the cylinder body. Then, when a cylinder is measured e.g. to calibrate or as a target, the first few minutes are usually discarded to make sure the regulator is flushed properly and to avoid such effects. This is recommended good practice in every WMO/GAW report. If a cylinder is used only sporadic, the regulator should be flushed several times anyway. Therefore, we do not think that this makes a big difference if best practice recommendation for trace gas measurements are followed properly.

The study was carried out with dry air mixtures ( $H_2O < 1 \mu mol mol^{-1}$ ). Residual water content might have a significant impact on the behavior of CO2 absorption. Has the low water content be verified by measurements or other means?

The water content is ensured by the drying system NOAA uses to fill the calibration gas tanks at Niwot Ridge station and is continually measured (Meeco Waterboy, Meeco Inc., USA) during the filling procedure. For further information such as the setup and specifications of the filling station and the procedure see: https://www.esrl.noaa.gov/gmd/ccl/ccl.html and https://www.esrl.noaa.gov/gmd/ccl/airstandard.html.

What will be the effect of residual water, even if less than 1  $\mu$ mol mol-1? Could it be that differences in the residual water content explain at least partly the difference between individual cylinders or fillings?

At such low water contents, we do not expect any significant influence of the water on the  $CO_2$  adsorption/desorption effects, because  $CO_2$  molecules occupy only a fraction of the available active sites on the cylinder wall as shown in the paper. It would be different if there were more water in the cylinder, because the water molecules would compete with the  $CO_2$  molecules for the available active sites and hinder  $CO_2$  molecules to adsorb to the cylinder wall.

A very recent publication studies similar effects including the influence of water on the stability of gaseous reference materials (Brewer et al., 2018). Citation of this work should be made in the final AMT version of the paper.

The publication was added to the references.

SGS (Superior Gas Stability) cylinders are mentioned in the introduction and methods, but no results are shown in the paper. In the conclusions, they are mentioned again, saying that they behave in the same way as untreated cylinders. This should also be shown and discussed in the results. E.g. individual fits could be shown for SGS and untreated cylinders in a separate figure similar to Fig.6.

We added the following sentence to section 3.2:

"The two SGS cylinders do not show a significantly different behavior, the form of the  $CO_2$  enrichment with decreasing pressure as well as the amount is the same as for the normal cylinders within the given uncertainty (Fig. 6 b)."

and we changed the sentence at page 10, line 34 from:

"Two additional low flow measurements with horizontally positioned cylinders were done."

to:

"Additionally, a low flow run with two horizontally positioned cylinders (one normal and one SGS cylinder) was done."

Additionally we expanded figure 6 that shows all low flow measurements of one cylinder (panel a) with a second panel that shows the average fit of all low flow measurements of the normal and SGS cylinders, respectively, with the uncertainties given as greyed area (panel b).

"Figure 6: a) The blue circles represent the  $CO_2$  mole fraction measurement of a low flow experiment started on 17.10.2016 with CB11873 vertically positioned as a function of pressure, note the inverse pressure scale. The black dashed lines are the individual fits following the Langmuir model of the other low flow experiments done with CB11873 vertically positioned, the black solid line represents the average Langmuir fit using all low flow experiments with the cylinders vertically positioned. b) The black solid and dashed line correspond to the average Langmuir fit of all normal and SGS cylinder measurement, respectively, that were done under low flow conditions, the greyed area corresponds to the standard deviation of the averages. In order to plot all data in one plot, the corresponding  $(CO_{2,ini} - CO_{2,ad})$  was subtracted from the measurements and the fits in both panels."

**In section 3.3, we replaced the first sentence**

"In the high flow mode, eight complete drainings were done with cylinders vertically positioned..."

with

"In high flow mode, each of the six normal and the two SGS cylinders were drained once with cylinders vertically positioned..."

In section 4, the first sentence

"The low flow measurements with cylinders vertically positioned show repeatedly comparable CO2 enrichment with decreasing pressure, no matter which cylinder was measured."

was shortened and a second sentence was added, it reads now:

"The low flow measurements with cylinders vertically positioned show repeatedly comparable  $CO_2$  enrichment with decreasing pressure. Neither the normal nor the SGS cylinders showed any unique features with respect to  $CO_2$  enrichment."

I don't see much additional value of the experiments with heating and changing the orientation of the cylinders during venting. The results of the heating experiments were not consistent between different runs, and probably more experiments would be needed to get a clear picture. For example, the mole fraction change after the start of the heating shown in Fig. 15 is not significantly different from changes observed at higher pressures during the same run. The results of the experiments with changing orientation during the draining are also based on only one run for each experiment, and it is unsure if they can be reproduced. The paper could be shortened and would improve if only the results of the low and high flow experiments, including different cylinder orientation and treatment, are presented.

We disagree with the referee that the heating experiments and changing the position do not add any value to the manuscript. However, we agree that more experiments are needed, which is also stated in the manuscript. This is also why we would like to keep these experiments in the manuscript. These tests with no clear result might inspire other laboratories to do more experiments in this direction to solve this issue. However, we shortened the paragraph at page 17, line 22 from

"With the three cylinders that were moved upside down, the picture is not very clear. Because of a logger failure, there are no temperature measurements on the first cylinder, which is why it will not be discussed here. The second and the third cylinders put upside down show a drop in the  $CO_2$  mole fraction after they have been moved. While the second cylinder remains stable at the slightly lower  $CO_2$  mole fraction, the  $CO_2$  mole fraction of the last cylinder goes up first by about 0.07µmol mol-1 from 30 to 15 bar and falls back by roughly the same amount until the end of the experiment. The temperature measurements of the two cylinders look the same. The only small difference between the two is the pressure when they were turned, the second cylinder was turned at 32.2 bar, the third cylinder was turned at 27.8 bar. However, whether this caused the different behavior in the  $CO_2$  measurements remains unclear. The results from the experiments with constant heating..."

to

"With the three cylinders that were moved upside down, the picture is not very clear. Also the results from the experiments with constant heating..."

I further recommend re-writing the conclusions. Currently, they are difficult to read without the full context of the paper, and present results which are not mentioned previously (e.g. SGS cylinders). Furthermore, the statement 'This opens the possibility to use a general correction

function in case a calibration cylinder on a field station runs empty' should be made in the results section because it needs more careful discussion. Most likely, corrections will be associated with high uncertainties, since the calibration sequence at stations is different from your experiments in which the cylinders were emptied with a constant flow.

**We changed the conclusion from:**

[revised manuscript text omitted]

In high flow settings (5.0 L min-1), additional thermal diffusion effects and Rayleigh fractionation come into play that add to, or can overrule the simultaneously ongoing Langmuir adsorption/desorption. Depending on the positioning of the cylinder, CO2 can be increasing or decreasing with decreasing pressure. We have demonstrated that these effects very likely do play a role, but before a satisfactory explanation can be attempted a considerable number of additional controlled experiments, as well as modeling of the flow and mixing in cylinders will be necessary. A further benefit could be gained by using a CRDS (cavity ring down spectroscopy) gas analyzer because it does not need to be calibrated as often as an NDIR analyzer and it could measure several gas species, such as CH4 or CO, simultaneously."

**Specific comments**

Page 2, line 13: kilogram is a SI unit, despite the fact that it is still based on an artifact. It should be removed from the list of examples in parenthesis.

```
"..., the kilogram or ... " was removed
```

Page 3, line 21ff: Add a short description of the performance of the analytical system (repeatability, drift etc.) here. This could be done by moving paragraph 3.1 to the method section.

We moved paragraph 3.1 to the method section and changed the paragraph and figure numbering accordingly.

Page 3, line 34: According to Fig. 1, C1 is repeatedly measured, not C2. Page 9, first lines of result section and section 3.1 would better fit in the method section.

Correct, changed to C1.

Page 10, section 3.2.: The low flow experiments are probably the most relevant for most users of standard gases. Beside the average of all cylinders, the result of only one (representative) standard is shown, while in total 38 experiments of the same type were made. It would be valuable to see the variation between different cylinders / fillings, which could be added in either an additional figure or Fig. 6 (e.g. individual fits for all experiments). I recommend to also show and discuss the similar behavior of SGS and untreated cylinders could here.

We added a second panel to Fig. 6, that shows the average fits of the low flow experiments of the normal and the SGS cylinders with the uncertainty given as greyed area (see also similar question above).

Figures 6 and 8: The y-axis shows  $\Delta CO_2$ , and not  $CO_2$  mole fraction, which needs to be corrected.

We replaced " $CO_2$  mole fraction..." with " $\triangle CO_2$ ..." in Figure 6 and 8. The same is true for Figure 12, which was changed in the same manner on the authors' behalf.

Why do the measurements at higher pressures show a negative delta? Especially in Fig. 8 all  $\Delta CO_2$  as well as the fits at higher pressures are negative. Is this correct?

 $(X_{CO2,ini} \text{ and } X_{CO2,ad} \text{ correspond to } CO_{2,ini} \text{ and } CO_{2,ad}, \text{ respectively, see also reply to Referee#2})$

It is correct but might be misleading. For this figures, the term  $CO_{2,ini}$  was subtracted from the Langmuir-equation (equation 1) and the remainder with the corresponding coefficients was plotted.  $X_{CO2,ini}$  corresponds to the  $CO_2$  mole fraction before adsorption to the walls occurs, therefore it is slightly higher than the  $CO_2$  average value at the beginning of the measurements. This results in negative  $\Delta CO_2$  values in the beginning of the measurements. For clarity we shifted the values by adding  $X_{CO2,ad}$ , this shifts the fits upwards so that the initial value is exactly 0. This was done for Fig 6, 8, and 12. In all three captions we changed

"...the corresponding CO2,ini was subtracted..."

to

"... the corresponding  $(X_{CO2,ini} - X_{CO2,ad})$  was subtracted...",

accordingly.

**Technical corrections**

Page 2, line 24: the latest available GGMT report is not cited (WMO, 2016). It should be added. The format of the citations of the WMO reports needs also to be changed.

The latest GGMT report was added and the format of the citations were changed.

Page 2, line 28: replace 'SI values' with 'SI traceable values'.

Done.

Page 2, line 35: Cite the latest GGMT report here.

Done.

Page 3, line 36: Change to 'An additional full calibration was made at the end of each experiment'.

Done.

Page 6, line 25: Kitzis (2017) is missing in the references.

Correct, we added the reference.

Corrections on the authors' behalf:

We replaced all symbols for liter "l" with a capital L for better readability (also in the figures).

As mentioned above, the y-axis of Fig. 12 was changed from " $CO_2$  mole fraction..." to " $\Delta CO_2$ ..." as requested by Referee#1 for Fig. 6 and Fig. 8.

Page 5 line 21: For consistency, we changed "normal flow" to "low flow"

Page 9, line 27: For clarity, we changed the first sentence from:

"No general filtering was applied to the measured data."

То

"No data selection was applied to the measured data."

Page 10, line 32 ff: We replaced the two sentences

"The standard deviation of the average  $\alpha$  is very low. If the measurement system's repeatability as deduced from the target gas measurements is taken into account, a realistic error of  $\alpha$  should be about four times bigger."

with

"The given uncertainty range of 0.000004 corresponds to about 9.3 % of (1-0.999957). Considering the calculated  $CO_2$  enrichment of 0.085 µmol mol-1, 9.3 % equates to about

0.008  $\mu$ mol mol-1, which is consistent with the measurement system's repeatability of 0.01  $\mu$ mol mol-1 as deduced from the target gas measurements before the analyzer change."

Page 13, line 6: For better readability the two sentences were shortened from

"If only the CO2 measurements below 50 % of the cylinder's pressure are used to calculate  $\alpha$ , then the average fractionation factor for the outflow becomes  $1.00021 \pm 0.00004$ , indicating an even stronger fractionation. When the Rayleigh fractionation with the stronger fractionation factor is only applied after the cylinder is half empty, when the temperature difference between the upper and the lower side of the cylinder body reaches its final value of 0.3 K, the average final depletion is  $-0.26 \pm 0.07 \,\mu$ mol mol-1."

to

"If only the CO2 measurements below 50 % of the cylinder's pressure are used to calculate  $\alpha$ , then the average fractionation factor for the outflow becomes  $1.00021 \pm 0.00004$ , indicating an even stronger fractionation with a final average depletion of  $-0.26 \pm 0.07 \mu mol mol^{-1}$ ."

Page 15 line 7: "vertically" was replaced with "horizontally".

Page 16, line 24: For clarity reasons we changed the sentence from

"This is also the moment when the temperature difference between..."

to

"This is also the reason when the measured temperature difference between..."

Page 18, line 18:We replaced 0.036  $\mu$ mol mol-1 with 0.034  $\mu$ mol mol-1

Page 24: For clarity reasons, the caption of figure 3 (now figure 2) was changed from

"Flow schematic of the high flow inlet system. The sample gas enters on the left side at 5.0 l min-1. A small aliquot of 0.3 l min-1 goes to the analyzer, the vast remainder of 4.7 l min-1 goes to the exhaust. The ratio between the gas going to the analyzer and the exhaust, respectively, can be adjusted by the needle valve on the exhaust side."

to

"Flow schematic of the high flow inlet system. The sample gas enters on the right side at 5.0 L min-1. A small aliquot of 0.3 L min-1 goes to the analyzer, the remainder of 4.7 L min-1 goes to the exhaust. The ratio between the gas going to the analyzer and the exhaust, respectively, is set by the dimensions of the inner and outer tube and can be adjusted by the needle valve on the exhaust side."

Page 28, line 4: "vertically" was replaced with "horizontally"

Page 34: We changed the colors of figure 13 for better readability and changed the caption accordingly.

Page 35: The x-axis title of figure 14 was corrected.

Page 37, line 5: "*vertically*" was replaced with "*horizontally*", the x-axis title of figure 16 was corrected.

---

## Author Comment (AC2) · 12 Jul 2018

The authors would like to thank Referee#1 and Referee#2 for their valuable comments and suggestions on this manuscript. Below we have addressed the individual remarks by each reviewer. The Referee's comments and questions are given in blue, the authors' replies are formatted as plain text, and excerpts from the manuscript as well as changes to the manuscript are given in italics.

Reply to Anonymous Referee#2

General comments:

This paper describes a series of experiments performed to better understand previously observed drifts in the $CO_2$ mole fraction measured during the lifetime of reference gases in high pressure cylinders. The subject is of interest for worldwide measurements of atmospheric $CO_2$ mole fractions, which make use of such reference gases for their calibration. The authors have planned a consistent number of experiments to study the impact of various conditions. The measurements themselves appear robust, and all plots clearly summarise the observations. There is however some issue with the overall presentation, the organisation of the results, the link with the equations developed to fit the measurement results, and the relevance of some of the experiments. I therefore recommend a major revision before the paper can be published in AMT.

Although the general aim of the study appears to be a better understanding of already observed effects when using $CO_2$ in air standards, it is not so clear which particular questions are answered. For example, it is already recommended to leave not less than 20 bars in those cylinders, and this recommendation is again confirmed here. If that was the goal, it should be stated.

It was not the goal of this study to confirm the 20 bar recommendation, however, since this threshold is confirmed by the measurements done for this study we thought it worth mentioning.

The study also looked at the influence of the flow rate at which the gas is used, but the conclusions are never actually linked with some recommendation. Does this have any impact on their usage? Similarly, a big part of the paper is devoted to the analysis of thermal convections inside the cylinders. What is the impact here? Should this be followed by any recommendation, or was it only performed to better understand the process?

The high flow experiments were done to reach a better understanding of the processes involved. As described in the introduction (page 3, line 8) the high flow measurements were done to find out whether adsorption/desorption processes are the only ones changing the $CO_2$ mole fraction of the sample gas with decreasing pressure. However, while doing these experiments, we found that Rayleigh fractionation is at work as well and that this effect is stronger than the Langmuir adsorption/desorption when decanting at high flow. Since we found this thermally induced fractionation process to play a major role in altering the $CO_2$ mole fraction at high flow rates, we wanted to investigate it further and be able to give a possible explanation for the measured effects as well.

The organisation, terminology and wording of the paper can be improved. The description of the experiment is not very well structured and information on a same aspect is sometimes separated in different sections. At several occasions, words have been omitted, resulting in sentences which can still be understood but belong to oral language rather than a written paper.

We replaced all contracted forms such as e.g. didn't, isn't etc. with their longer forms (did not, is not etc.). We also changed phrases throughout the whole manuscript to improve readability, e.g. page 7, line 15 *"...where the $CO_2$ increase is more pronounced..."* reads now *"...where the increase of the $CO_2$ mole fraction is more pronounce*d…*"*

Authors also chose to sometimes use their own terminology, not following recommendations from international bodies such as IUPAC. It would be easier to follow with a more common terminology. It is also suggested to write all symbols for quantities in italic.

All symbols of quantities are now in italic and we changed some of the symbols used in the manuscript (e.g. $CO_{2,ad}$ is now $X_{CO2,ad}$, $N_a$ was changed to $n_a$ in order to prevent confusion with the Avogadro number, l (liter) was changed to L etc.)

Finally, a recent paper dealing with the same subject was published earlier in 2018. It should be mentioned in the introduction and results compared in the discussion: Brewer, P. J., R. J. C. Brown, K. V. Resner, R. E. Hill-Pearce, D. R. Worton, N. D. C. Allen, K. C. Blakley, D. Benucci and M. R. Ellison (2018). "Influence of Pressure on the Composition of Gaseous Reference Materials." Analytical Chemistry 90(5): 3490-3495.

We added this paper to the references.

Specific comments by section:

1. Introduction: the end of that section needs to be revised to state more clearly which questions are being answered, which experiments were performed to do so, and which theory was applied. It is suggested to replace the assumptions with questions, such as i) is the effect cylinder−dependent? ii) in which conditions is the Langmuir model sufficient, and when does this need to be completed with the other effects already mentioned earlier in the section? iii) are SGS cylinders any better?

We changed the wording of our hypotheses from

*"In this study the following hypotheses were tested, i) the $CO_2$ increase with decreasing pressure is different for each individual cylinder, ii) the $CO_2$ enrichment follows the Langmuir monolayer adsorption/desorption model and iii) the stability of the $CO_2$ mole fraction is better in SGS (Superior Gas Stability®, Luxfer, USA) cylinders."*

to

*"In this study, three hypotheses were tested, i) the increase of the $CO_2$ mole fraction in the sample gas with decreasing pressure is different for each individual cylinder, ii) at low flow rates, the Langmuir monolayer adsorption/desorption model is sufficient to describe the observed $CO_2$ enrichment in the sample gas with decreasing pressure and iii) the stability of*

*the $CO_2$ mole fraction with decreasing pressure is better in SGS (Superior Gas Stability®, Luxfer, USA) cylinders than in untreated aluminum cylinders.*"

In addition, other questions may be added if there are parts of the goal of the study, such as the influence of the flow rate at which the gas is sampled from cylinders, or the influence of the cylinder position.

Since most measurement systems use calibration gases at low flow rates, our focus is on the low flow rates. It was not the goal of this study to show that different flow rates as well as changing the orientation alter the $CO_2$ mole fraction of the sample gas. We used the different flow rates and orientations to see whether adsorption/desorption processes are the only effects influencing the $CO_2$ mole fraction of the sample gas or whether temperature gradients and thereby induced fractionation also plays a role. This is also described briefly on page 3, line 10 ff. However, since we found that a high flow rate in combination with different orientations of the cylinders has an influence on the $CO_2$ mole fraction of the sample gas, we thought it worth mentioning.

To make it more clear that we will ad change hypothesis two from

"*…ii) the $CO_2$ enrichment follows the Langmuir monolayer adsorption/desorption model…*"

to

"*…ii) at low flow rates, the Langmuir monolayer adsorption/desorption model is sufficient to describe the observed $CO_2$ enrichment with decreasing pressure…*"

Finally, it is recommended to add a summary of the following sections, to clarify for the reader which information is provided in each section.

We think by rearranging the sections as recommended by Referee#1 and Referee#2 the manuscript should be much easier to follow and therefore we do not think that this is necessary anymore.

2. Section 2:

a. The title ("methods") appears unclear. It actually contains information on the cylinders and analytical equipment, the measurement protocol, but also some theory to describe all processes at work.

We changed it to "*Material and methods*"

To our understanding the purpose of the methods section is to describe the used materials, the analytical equipment, the measurement protocol, how the data was analyzed and what type of fit functions were used and why.

b. It is suggested to modify the organisation, starting with a section only devoted to the description of the theory, then the equipment (cylinders plus gas lines plus instruments), and finally the measurement protocol. Within the protocol, the description of the measurement

sequence can come first, followed by the explanation of the different conditions chosen to test the various assumptions.

We reorganized the method section to the following order: Sample cylinders, Measurement system with subsections protocol, low flow measurement and high flow measurement, system performance, temperature measurement, heating system, $CO_2$ enrichment estimates.

2.1 Sample cylinders

2.2 Measurement system

2.2.1 Low flow measurements

2.2.2 High flow measurements

2.2.3 Measurement protocol (with explanation of the different conditions to test the various assumptions)

2.2.4 System performance

2.3 Auxiliary systems

2.3.1 Temperature measurement

2.3.2 Heating system

2.4 $CO_2$ enrichment estimates

2.4.1 Langmuir adsorption/desorption model I

2.4.2 Langmuir adsorption/desorption model II

2.4.3 Estimating $K$ based on the $CO_2$ measurements

2.4.4 Rayleigh distillation model and its combination with the Langmuir adsorption/desorption model I

The reasoning behind this order is: We are interested in the stability of $CO_2$-in-dry-air mixtures in high pressure aluminum cylinders at low flow rates, therefore the cylinders should be described first, followed by the sections where we show how we measured it and how good the performance of the measurement system is. After that, auxiliary systems (temperature measurements of the cylinders and heating) are described. The last step of this work, and therefore presented last, are the different functions of known effects we used to fit to the measured data, to find out which effect is most probably responsible for the observed $CO_2$ enrichment.

c. The description of the protocol needs to be revised. Table 1 and figure 1 are clear and almost describe it completely, but the text brings more confusion. Specific words need to be chosen for each series of measurement (block, cycle, run), defined and further used with always the same meaning.

We changed the description of the measurement cycle from

*"The calibrations and measurements were done in a cyclic sequence defined in the control program. The cycles were divided in blocks of 5 minutes each, during which gas from a single cylinder (sample, calibration or target gas) was measured. The data was read every 5 seconds, but data reported to the log file were 30 s averages yielding 10 values per block. In the very beginning of each cycle, a full calibration with all four calibration gases (C1, C2, C3, C4) as well as a target gas measurement were done. Then the program cycled through all samples several times, before C1 was measured again to catch short term drifts in the measurement system (Fig. 1)."*

to

*"The $CO_2$-analyzer reported 5 s values to the data logger, which in turn logged 30 s averages. Ten of these 30 s averages were taken together into one 5 minute block that formed the basic unit used for the measurement sequence. The calibration, target and sample gas measurements were done in a repetitive cyclic sequence that was made up of the aforementioned 5 minute blocks and whose order was defined in the control program. In the very beginning of each cycle, a full calibration with a single block of each calibration gas (C1, C2, C3, C4) as well as a block of target gas measurement were done. Then, the program switched through all connected samples several times, measuring a block C1 in between to catch short-term drifts of the measurement system (Fig. 3). When a cycle was finished, a new cycle was started, again by measuring blocks of all four calibration gases first. At the end an additional calibration with all four calibration gases and the target gas was made."*

A block is a 5 min interval during which a single gas is measured, the measurement cycles are made up of blocks.

A cycle starts by measuring all calibration gases, each during a 5 minute block, and subsequent sample gas measurements and some blocks of C1 to catch short term drifts of the analyzer.

A "cylinder measurement" refers to all measured blocks of one individual cylinder during one run.

During a run, one or several sample cylinders are drained from their filling pressure down to the preset pressure threshold, when the measurement is stopped (usually 1 or 1.5 bar)

d. First part of the section 2 describes the measurement sequence, also displayed in Figure 1. Sequences were apparently setup to obtain both calibrated and a drift corrected values. This could be stated first, before providing the details of how this goal was achieved.

We changed the wording, see previous reply.

3. Section 2.4: is it relevant to show the equation used to determine the temperature? This cannot be checked and the only interesting value in that case seems to be the temperature.

The equation was removed and the sentence

*"The voltage was converted into temperature values by using the Steinhart-Hart (Steinhart and Hurt, 1968):*

$$\frac{1}{T} = A + B \cdot \ln R + C \cdot (\ln R)^3 \tag{1}$$

*where T is the Temperature, A, B and C are the Steinhart-Hart coefficients provided by the manufacturer of the thermistors and R is the measured resistance."*

reads now

*"The voltage was converted into temperature values by using the Steinhart-Hart equation (Steinhart and Hart, 1968)."*

4. Section 2.6: it is indicated that the filling was done at Niwot Ridge Station. Was this the case every time a cylinder was emptied? Was it sent back and refilled there? Please clarify.

Yes, it was always the case. Niwot Ridge Station is very close to the NOAA Campus in Boulder. It is run by our group, which is why we filled the cylinders always there using the same equipment and procedure as for the NOAA standard cylinders.

5. Section 2.7: the entire section is difficult to follow, partly due to some considerations on unsuccessful attempt to fit the data. Even if some choices were justified by the observations, it would be easier to read if the different models were presented independently. The section could be divided in three parts, each of them presenting the assumptions, which part of the information comes from previous work (seems to be all in model 1 for example), which is the equation used during the fits, which parameters were assumed and which were fitted. The presentation of the maths could be better balanced: while some common knowledge is sometimes detailed (such as the definition of the Avogadro number), some more expert information is not fully described (such as the Rayleigh distillation function).

We merged other sections and split this section in four parts, it is now:

2.4 $CO_2$ enrichment estimates

2.4.1 Langmuir adsorption/desorption model I

2.4.2 Langmuir adsorption/desorption model II

2.4.3 Estimating $K$ based on the $CO_2$ measurements

2.4.4 Rayleigh distillation model and its combination with the Langmuir adsorption/desorption model I

There we moved the explanations about the difficulties of fitting the K value into the according section, which reads:

**"2.4.3 Estimating K based on the $CO_2$ measurements**

"To find a value for *K*, a process of elimination was used. Given that the residuals between the data and the fit function of a good fit are normally distributed, *K* can be found by fitting the adsorption/desorption equation but with a fixed *K* value, starting with a value close to 0. Then, *K* is increased step wise, until the residuals are not normally distributed anymore. To improve the sensitivity of this method, only $CO_2$ measurements below 30 bar were taken into account, where the increase of the $CO_2$ mole fraction is more pronounced. These calculations were done for ten different low flow cylinder measurements. The resulting *K* values were averaged and the standard deviation was calculated. This *K* value might not be the best fit, but it gives a good estimate about the upper boundary of possible K values. To make sure the residuals of all fits stay well within the normally distributed range, the *K* value was considered to be the difference "average *K* value minus standard deviation", which resulted in 0.002 bar$^{-1}$. To be on the safe side, *K* was set in the nls algorithm to 0.001 bar$^{-1}$."

a. Model 1: Langmuir. Equation (2) comes from Leuenberger 2015 where it was derived from the Langmuir model. This should be stated clearly.

We changed the sentence

"*...based on the Langmuir adsorption/desorption model (Langmuir, 1918, 1916; Leuenberger, 2015):...*"

to

"*...based on the Langmuir adsorption/desorption model (Langmuir, 1918, 1916) as derived by Leuenberger et al. (2015):...*"

In addition, the Langmuir model normally uses partial pressure, not total pressure. Some consideration on the choice of using total pressure and its impact should be provided. The explanation on the difficult fitting may be moved to the results, to limit this section to the theory. The quantity K was fixed at the same value than in the Leuenberger paper. Is that a coincidence? This should be better explained, but again not in this section, which should only state that K was fixed to find $CO_2$,ad.

The assumption that the $CO_2$ partial pressure is proportional to the total pressure is valid in our case because the $CO_2$ mole fraction varies by at most a few tenths of 1 ppm out of 400 ppm. Method II does not make that assumption.

We decided against moving the explanation to the results section but we split the section in four subsections, one of these subsections explains how K was determined.

We shortened the following sentence

"*Therefore the algorithm wasn't able to find K values with a high confidence level and the output corresponded mostly to the lower boundary, even when it was set to 0, meaning no exchange between the cylinder wall and the gas.*"

to

*"Therefore the algorithm was not able to find K values with a high confidence level and ultimately K was fixed at 0.001 bar$^{-1}$ to find the other coefficients of the model, as will be explained later."*

The rest of the explanation will follow in a separate section "*2.4.3 Estimating K based on the CO$_2$ measurements*". The content of this new section is given above at comment 5.

It is true that K was fixed at the same value as in Leuenberger et al. (2015) but the K value was estimated independently. However, the value used here only represents a rough estimate and was fixed for all the further calculations.

b. Model 2: in this approach, it is stated that the fraction of occupied sites does not depend on the total pressure. Note that the CO$_2$ mole fraction varies with the total pressure, as demonstrated by the paper. It is certainly negligible but this should be explained. This section introduces the symbol Na for a number of moles. This symbol being commonly chosen for the Avogadro number, it is recommended to replace with na. The derivation of Equation (9) from (8) is not straightforward, compared to equation (5) for example which is very straightforward. It is suggested to provide the steps in annex, limiting this section to the model description and the equation used later on to fit the data. The formalism could also be improved, avoiding a sentence such as "total trace gas" to replace a quantity. Why not defining ntot = nad+ngas. Finally, like for the first model, considerations on results of the fit should be kept for the results analysis.

To make it easier to follow the derivation of equation 9, we added the following steps between Eq. (8) and Eq. 9 and changed the numbering of the equations accordingly:

"*...which we rearranged into the following equation*

$$-\rho_a dX = \frac{a}{V}\left(\frac{Kd\rho_x}{1+K\rho_x} - \frac{K\rho_x Kd\rho_x}{(1+K\rho_x)^2}\right) = \frac{a}{V}\frac{Kd\rho_x}{1+K\rho_x}\left(1 - \frac{K\rho_x}{1+K\rho_x}\right) \tag{9}$$

*and subsequently into*

$$-\rho_a dX = d\rho_x \frac{a}{V}\frac{K}{(1+K\rho_x)^2} \tag{10}$$

*By substituting $d\rho_x$ in Eq. (10) with $Xd\rho_a + \rho dX$, it can be rearranged to...*"

We added "$n_{tot} = n_{ad}+n_{gas}$" as equation 5 and replaced the term "*Total trace gas*" with "$n_{tot}$" in the other formulas accordingly. "$N_a$" was replaced with "$n_a$" as suggested.

c. Model 3: in the third part, a Rayleigh distillation function is introduced. Some more explanation would be needed here, describing in one sentence the process and its expected impact on the CO$_2$ mole fraction. The formalism may also be improved here: equation (12) introduces a mole fraction with the symbol X, where equation (13) uses CO$_2$, meas. It is suggested to always use x for mole fractions.

The sentence

"*The outflowing gas is depleted in $CO_2$ if $\alpha < 1$ and vice versa.*"

was changed to

"*The outflowing gas is depleted in $CO_2$ if $\alpha < 1$, leaving the gas in the cylinder slightly enriched in $CO_2$ (and vice versa). With ongoing outflow, the effect gets stronger because the gas in the cylinder becomes more and more enriched. However, to make Rayleigh distillation possible, a fractionating process has to be involved, namely that the $CO_2$ mole fraction of outflowing gas is either enriched or depleted with respect to the cylinder average. A possible reason for fractionation in the cylinders is a temperature gradient in the cylinder. Heavier molecules tend to accumulate at the cooler end of a gas reservoir, while the lighter molecules are slightly more represented at the warmer end. If the sample air is taken from e.g. the warmer part of the gas column, it will be slightly depleted in the heavier molecule while the gas in the cylinder becomes enriched.*"

We replaced "$CO_{2,meas}$" with "$X_{CO2,meas}$"

6. Section 3:

a. This section is quite long and it is not always easy to distinguish the conclusions that are derived from each particular experiment. It is suggested to revise the structure, splitting in sub− sections to clearly identify what is the tested assumption, what were the observations, and the preliminary conclusion.

We split section 3.2 High flow measurements (formerly 3.3, see reply to 7) and merged it with "Moving cylinders into different orientations while measuring" and the "Heating cylinders" section, it is now:

3.2 High flow measurements

3.2.1 Vertically positioned cylinders

3.2.2 Horizontally positioned cylinders

3.2.3 Moving cylinders into different orientations while measuring

3.2.4 Heating cylinders

To clarify the tested assumption we changed:

 The first sentence of Section 3.1 Low flow measurements (formerly 3.2) from

"*In the low flow mode, 38 full tanks were depleted with vertically positioned cylinders.*"

to

"*In the low flow mode, 38 full tanks were depleted with vertically positioned cylinders to see whether the $CO_2$ mole fraction change with decreasing pressure is different in each individual cylinder and whether SGS cylinders perform better than normal cylinders."*

At Section 3.2 High flow measurements (formerly 3.3) we changed the first sentence from

"*In the high flow mode, eight complete drainings were done with cylinders vertically positioned.*"

to

"*In high flow mode, each of the six normal and the two SGS cylinders were drained once with cylinders vertically positioned, to find out whether Langmuir adsorption/desorption is the only process enriching the $CO_2$ mole fraction with decreasing pressure.*"

At page 11, line 34 we changed the sentence from

"*Three more complete drainings were done with the cylinders horizontally positioned.*"

to

"*Three more complete drainings were done with the cylinders horizontally positioned to measure the $CO_2$ changes with different temperature gradients compared to the vertically positioned cylinders. In case the temperature gradient has no influence on the observed $CO_2$ changes with decreasing cylinder pressure, the outcome of these measurements should be the same as with vertically positioned cylinders.*"

At Page 13, line 13 we added a sentence, it reads now

"*...at about 30 bar and vice versa. If there are different air masses in the cylinder with different temperatures and therefore depleted/enriched $CO_2$ mole fractions, moving the cylinder should cause a sudden change in the temperature measurements as well as in the $CO_2$ measurements of the sample gas. The tank that was first...*"

At page 14, line 7 we added a sentence, it reads now

"*...at a cylinder pressure of 30 bar. In case the heating induces convection in the cylinder, we expect the sample gas to become well mixed and no $CO_2$ enrichment besides Langmuir adsorption/desorption as in the low flow. The first cylinder showed...*"

To emphasize the preliminary conclusions we changed:

Page 10, line 22ff from

"*A fit following Langmuir's adsorption/desorption model was calculated for each measurement. The fit functions were used to calculate the average $CO_2$ enrichment with decreasing pressure using the pressure measurements, it is $0.089 \pm 0.013$ μmol $mol^{-1}$ (Fig. 6), the given error corresponds to the standard error (1-sigma) of the average. Since each cylinder started with a different pressure, the averages could be not entirely comparable. However, if the enrichment is calculated over the same pressure span of 150 to 1 bar, the result is again $0.090 \pm 0.009$ μmol $mol^{-1}$, which is the same within the given uncertainty.*"

to

*"A fit following Langmuir's adsorption/desorption model was calculated for each measurement and used to estimate the average $CO_2$ enrichment with decreasing pressure. Using the actual pressure measurements, the average $CO_2$ enrichment is 0.089 ± 0.013 μmol mol$^{-1}$ (Fig.6 a), the given error corresponds to the standard deviation (1-sigma) of individual cylinder drainings. However, values for the $CO_2$ enrichment of the individual cylinder measurements might be not entirely comparable, since each cylinder had a different initial pressure. Therefore, we calculated the $CO_2$ enrichment for each cylinder measurement using the same pressure span of 150 to 1 bar. This results in an average $CO_2$ enrichment of 0.090 ± 0.009 μmol mol$^{-1}$, which is the same within the given uncertainty. The variation of the enrichment was very low, indicating that the $CO_2$ enrichment with decreasing pressure is not cylinder dependent. The two SGS cylinders do not show a significantly different behavior, the shape of the $CO_2$ enrichment with decreasing pressure as well as the amount is the same as for the normal cylinders within the given uncertainty (Fig. 6 b)."*

7. Section 3.1: there are many details in this section which may not be relevant for the understanding of the measurements. If the change of the analyser was not an issue, this should not be highlighted so strongly. The term accuracy does not seem appropriate, as the analyser was regularly calibrated. In addition it is noted at the end of the section that only relative measurements were performed. Then why not starting with this statement, and explaining that the repeatability (expressed with the standard deviation) of the measurements was evaluated, as this is the quantity which matters.

Following the recommendation of referee#1, this section was moved to the methods part and renamed system performance.

The change of the analyzer is not a big issue, however, it's still visible in the histogram and we wanted to explain, why the histogram shows two peaks with the second one having a higher variation.

In our opinion, the statement about the measurement performance fits better at the end of the section.

8. Section 3.2: it is mentioned here that the values $CO_{2,ini}$ and $CO_{2,ad}$ had a fix value. This was not so clear in the section on methods. It was also not clear how their values were actually chosen.

In the low flow experiments, $CO_{2,ini}$ and $CO_{2,ad}$ are not fixed, only K was fixed as mentioned in the method section. In section 3.2 (now 3.1) we state that the values we found for $CO_{2,ad}$ are almost the same in all low flow experiments but they were calculated for each run of each cylinder separately (page 10, line 27).

9. Section 3.3: it is suggested to separate this section in two, to treat vertical and horizontal cylinders separately. Also the reference to figure 9 appears too early, as this figure contains results with both positions of the cylinders. The clear distinction between them helps the understanding of the analysis, but maybe this could come after. Some effort to reduce the text, summarising each observation and drawing conclusions by step would be very much appreciated as well.

We split section 3.2 High flow measurements (formerly 3.3) and merged it with "Moving cylinders into different orientations while measuring" and the "Heating cylinders" section, it is now:

3.2 High flow measurements

3.2.1 Vertically positioned cylinders

3.2.2 Horizontally positioned cylinders

3.2.3 Moving cylinders into different orientations while measuring

3.2.4 Heating cylinders

10. Section 3.4: please indicate which assumption was tested here and also when the cylinders were put upside−down. Certainly the idea of moving cylinders came from the observations during previous measurements, but this is not very clear.

The respective pressures when the cylinders were moved can be found on page 13, line 32, line 34 and line 36 respectively.

At page 13, line 13 We added the following sentence to clarify why these experiments were done:

"*If there are different air masses in the cylinder with different temperatures and therefore depleted/enriched $CO_2$ mole fractions, moving the cylinder should cause a sudden change in the temperature measurements as well as in the $CO_2$ measurements of the sample gas.*"

11. Section 3.5: Some effort to reduce the text, summarising each observation and drawing conclusions by step would be very much appreciated as well here.

We shortened a few sentences and added a brief conclusion after the experiments with the constant heating as well as after the experiments with the burst heating.

Page 14, line 8ff was shortened from

"*The first cylinder showed a stable $CO_2$ mole fraction before heating, it was $410.36 \pm 0.03$ $\mu mol\ mol^{-1}$. The resumed $CO_2$ measurement after the cylinder temperature reached $30\ °C$ showed a drop of about $0.09\ \mu mol\ mol^{-1}$ and remained stable at $410.27 \pm 0.02\ \mu mol\ mol^{-1}$ until the cylinder was empty (Fig. 15). The second cylinder showed no changes in the $CO_2$ mole fraction before and during the heating. The average $CO_2$ mole fraction down to 30 bar was $410.99 \pm 0.03\ \mu mol\ mol^{-1}$, after the temperature reached the preset value it was at $410.99 \pm 0.02\ \mu mol\ mol^{-1}$.*"

to

"*The mole fraction of the first cylinder was stable before and after the heating was started at $410.36 \pm 0.03\ \mu mol\ mol^{-1}$ and $410.27 \pm 0.02\ \mu mol\ mol^{-1}$, respectively (Fig. 15). The second cylinder showed no changes in the $CO_2$ mole fraction before and during the heating. The*

*average CO₂ mole fraction coming out of the cylinder before and after heating was $410.99 \pm 0.03$ μmol mol⁻¹and $410.99 \pm 0.02$ μmol mol⁻¹, respectively."*

Page 14, line 18ff was shortened from

*"The cylinder wall is slightly thicker at the shoulder resulting in a bigger thermal mass. That, and the fact that there was a heat band at the bottom, might be why the temperature is higher at the bottom than at the shoulder and why the temperature overshot in the beginning of the heating."*

to

*"A possible reason for the cylinder being warmer at the bottom end might be the thicker wall at the shoulder that results in a bigger thermal mass and the additional heat band at the bottom of the cylinder."*

Page 14, line 33ff was shortened from

*"Then, after the heating was switched on, the temperature gradient was turned upside down almost immediately. The set temperature of 30 °C was reached after about one hour, which corresponds to a pressure decrease of about 10 bar."*

to

*"After the heating was switched on, the temperature gradient was inverted almost immediately and after about one hour the set temperature was reached."*

Two brief conclusions were added at page 14, line 20

*"Due to the inconsistency of the two runs, it is impossible to tell whether the mixing induced by convection prevented thermal fractionation or not."*

and at page 15, line 2

*"The two experiments showed, that quick heat bursts have only a short effect on the sampling gas and are not sufficient to produce much mixing the cylinder gas."*

12. Section 4: it is suggested to also reshape this section, starting from each conclusion drawn during section 3, and summarising. At the end, some consideration on the impact should be given. For example, a possible small bias in the calibration of CO₂ measurement is mentioned. What is the conclusion? Will it be followed by further recommendation?

We split the section in

4 Discussion

4.1 Low flow measurements

4.2 High flow measurements

On page 15, line 29 we changed the sentence

"*This effect is likely worse with smaller cylinders, where the surface to volume ratio is bigger*"

to

"*This effect is likely worse with smaller cylinders, where the surface to volume ratio is bigger and should be taken into account when preparing $CO_2$ standards gravimetrically or when preparing mother/daughter cylinder sets for comparison projects between different analyzing systems and/or laboratories.*"

At the end of page 15, it seems that some inconsistency is noted between the observations made when sampling the gas with low and high flow rates, as they would lead to two different values of the amount of $CO_2$ adsorbed on the wall. This appears quite serious. What is the conclusion? Does this question the entire model?

At page 15, line 30 ff, we state that the observed $CO_2$ enrichment in vertically positioned cylinders in the high flow setup is about 2.5 times higher compared to vertically positioned cylinders in the low flow setup. In the same paragraph we explain that the reason for this higher enrichment can be attributed to Rayleigh distillation processes, which do not occur under low flow conditions. Since most gas measurement systems have a low calibration gas flow, or if they have a high calibration gas flow, it is only for a few minutes, too short to induce a temperature gradient, this shouldn't be a problem.

At page 15, line 30 we changed the sentence from

"*…in the high flow measurement was on average…*"

to

"*…in the high flow measurement with vertically positioned cylinders was on average…*" for clarity reasons.

This is also stated as a conclusion in section "5. Conclusions", page 18, line 20 ff.

13. Section 5: the first conclusion from the low flow experiment appears to diminish the importance of the study. Indeed, if it is already recommended to stop using calibration gases when the pressure is below 20 bar, what was the purpose of this study? The second part would need to be strengthened; bringing more sounds conclusions and consideration on the implication (if any) on the usage of calibration gases.

At page 18, line 9 to 13 we verify/falsify the three hypotheses from the introduction. These are the most important findings of this study. The most important implications are given at line 13-15 (possibility to correct if necessary) and that recommendation from the WMO, to change cylinders when they reach 20 bar should be changed, still holds true, even though we recommend to change the cylinders when they reach 30 bar to minimize the effect of the enrichment.

Additionally, we added the sentence

*"By using bigger cylinders (e.g. 50 l) the surface to volume ratio becomes smaller compared to the 29.5 l cylinders used in this study, which might be beneficial in minimizing the $CO_2$ enrichment effect at lower pressures."*

at page 18, line 19.

The whole rest of this paragraph are additional findings we didn't look actively for but thought worth mentioning.

Line-by-line comments:

Line 20: "…In this study we found that during low flow conditions". As this is the first mention of low flow rates, it should be stated more clearly that this is the flow rate used to sample the gas inside the standard.

We replaced the sentence

*"In this study we found that during low flow conditions ($0.3\,l\,min^{-1}$) the tested vertically positioned aluminum cylinders always showed similar $CO_2$ enrichment of $0.090 \pm 0.009\,\mu mol\,mol$-1 as the cylinder was emptied from about 140 to 1 bar above atmosphere, following Langmuir's adsorption/desorption model."*

with the following sentence

*"In this study the tested vertically positioned aluminum cylinders showed similar $CO_2$ enrichment during low flow conditions ($0.3\,l\,min^{-1}$), which are similar to flows often used for calibration gases in practical applications. The average $CO_2$ enrichment was $0.090 \pm 0.009\,\mu mol\,mol^{-1}$ as the cylinder was emptied from about 150 to 1 bar above atmosphere. However, it is important to note that the enrichment is not linear but follows Langmuir's adsorption/desorption model, where the $CO_2$ enrichment is almost negligible at high pressures but much more pronounced at low pressures."*

Note that we corrected 140 to 150 bar, which is the pressure the calculations were based on. At the same time, this change should also address the next remark of Referee#2.

Line 21: "showed similar $CO_2$ enrichment of $0.090 \pm 0.009$ µmol mol$^{-1}$ as the cylinder was emptied from about 140 to 1 bar above atmosphere". This is misleading, because the increase happens only after a certain pressure, not gradually during the cylinder was emptied. This is quite important as users could be afraid of using the standards.

We disagree, according to Langmuir's adsorption/desorption model, the $CO_2$ starts to desorb as soon as the pressure starts to decrease. However, in the beginning (at high pressures) the number of $CO_2$ molecules desorbing from the walls is very small compared to the number of $CO_2$ molecules present in the cylinder air. At the end of the sentence, we state that the enrichment follows the Langmuir model, which should be a clear indicator, that the $CO_2$ enrichment does not follow a linear function.

This is the exact reason why the WMO states in its GAW reports: "In the case of $CO_2$ the calibration standards should be replaced once the cylinder pressure has decreased to 20 bar.".

(e.g. WMO: 18th WMO/IAEA Meeting of Experts on Carbon Dioxide Concentration and Related Tracers Measurement Techniques (GGMT-2015), La Jolla, CA, USA, 13-17 September 2015, GAW Report No. 229, World Meteorological Organization, Geneva, Switzerland, 2016.)

Line 28: "In case they are used in high flow experiments that involve significant cylinder temperature changes, special attention has to be paid to possible fractionation effects". This is the only statement in the entire paper about possible impacts of sampling gas from the standards at a high flow rate. Some more consideration should be provided, both in the introduction to explain the current situation, and in the conclusion to provide a recommendation.

The high flow experiments were only done to understand the involved processes better. For most applications, the low flow experiments are much more important. However, since we found these effects, we thought it worth mentioning, in case somebody actually uses cylinders at such high flows over a period long enough to induce temperature gradients.

The possible impacts are mentioned in the discussion page 15, line 30 ff, where we explain what the possible reason for this observation is, as well as in the conclusions page 18, line 20 ff.

Line 14-17: the sentence about the traceability explains the situation within WMO/GAW. It may not be extended to all $CO_2$ measurements in the world.

We changed the sentence from

"*In the case of GHGs traceability is maintained by the use of a unique hierarchy of $CO_2$-in-(dry)-air mixtures (and similarly for $CH_4$, $N_2O$) in high pressure cylinders starting from the primary standards (with link to SI) to secondaries and tertiaries, all with known $CO_2$ mole fraction derived from the higher level, ultimately calibrating the instrument making air measurements.*"

to

"*Within the WMO/GAW network, GHGs traceability is maintained by the use of a unique hierarchy of $CO_2$-in-(dry)-air mixtures (and similarly for $CH_4$, $N_2O$) in high pressure cylinders. The hierarchy starts from the primary standards (with link to SI) to secondaries and tertiaries, all with known $CO_2$ mole fraction derived from the higher level, ultimately calibrating the instrument making air measurements.*"

Line 18:"The resulting…" this sentence describes a goal rather than an observation. In addition the word "true" should be avoided and could be replaced with "unbiased".

Gas analyzers with well known characteristics should indeed not depend on priori estimates or models. A suite of calibration gases that covers the range in which the analyzer will be used is usually enough. Of course, this is different for e.g. remote sensing data such as satellite data, FTIR systems measuring the total column etc., these systems depend on modelling and a priori estimates of the column.

As suggested, we replaced "*true*" with "*unbiased*".

Line 28:" SI values". Incorrect terminology. May be replaced with "values traceable to the SI".

We changed "*SI values*" to "*SI traceable values*"

Line 32:"accuracy". Incorrect terminology. To be replaced with Data Quality Objective or compatibility.

"accuracy" was replaced with "compatibility"

Line 17: "To check the third hypothesis, two SGS cylinders were added to the set…" this would mean that 8 + 2 = 10 cylinders were tested. Table 1 shows only 8 cylinders (not considering the one which was replaced).

For clarity reasons, the sentence was changed from

"*To check the third hypothesis, two SGS cylinders were added to the set but used exactly the same way as the ordinary cylinders.*"

To

"*To check the third hypothesis, two of the eight tested cylinders were SGS cylinders, but they were used exactly the same way as the ordinary cylinders.*"

Line 30 and 32: the term "block" seems to indicate 10 measurements recorded during 5 minutes. Later in the text, the same quantity is sometimes only referred to as "a measurement". It can be understood to define a measurement as a 5 minute average, but this should be stated and always used in the same way.

For consistency reasons we applied the following changes:

Page 4, line 16, we replaced "…*and the target gas...*" with "…*a block of target gas measurements...*"

Page 4, line 20, we replaced "*measurement*" with "*run*"

Page 4, line 29, we changed "*...resulted in about 12 additional target gas measurements...*" to "*...resulted in about 12 additional blocks of target gas measurements...*"

Page 5, line 2, we changed "*...about 4 target gas measurements...*" to "*...about 4 blocks of target gas measurements...*"

Page 10, line 2, we changed "*...to the measurements...*" to "*...to the measurement sequence...*"

Page 10, line 5, we changed "*...were calculated for each target gas measurement...*" to "*...were calculated for each block of target gas measurement...*"

Page 10, line 20, we changed "*...All measurements followed a similar pattern...*" to "*...All low flow measurements followed a similar pattern...*"

Page 10, line 22, we changed "*...for each measurement...*" to "*...for each cylinder measurement...*"

Page 10, line 22, we changed the "*Two additional low flow measurements with horizontally positioned cylinders were done. Again the Langmuir fit functions of the two measurements were used...*" to "*Additionally, a low flow run with two horizontally positioned cylinders (one normal and one SGS cylinder) was done. Again the Langmuir fit functions of the two cylinder measurements were used...*"

Page 11, line 4, we changed "*...at the same time the only low flow measurement with temperature...*" to "*...at the same time the only low flow run with temperature...*"

Page 12, line 36, we changed "*...until the end of the measurement, the $CO_2$...*" to "*...until the end of the run, the $CO_2$...*"

Page 13, line 4: We changed "*In all three measurements the logarithmic plots show a flat plateau in the beginning. The decrease starts in all three measurements at $-ln(P/P_0) \approx 0.7$...*" to "*In all three runs the logarithmic plots show a flat plateau in the beginning. The decrease starts in all three runs at $-ln(P/P_0) \approx 0.7$...*"

Page 13, line 20, we changed "*...the two measurements seem to mirror...*" to "*...the two cylinder measurements seem to mirror...*"

Page 14, line 27, we changed "*In the second measurement, the $CO_2$ mole fraction...*" to "*In the second run, the $CO_2$ mole fraction...*"

Page 16, line 10, we replaced "*...at the end of the measurement.*" with "*...at the end of the run.*"

Page 16, line 26, we replaced "*...which is maintained until the end of measurement.*" with "*...which is maintained until the end of run.*"

Page 17, line 4, we replaced "*At the end of the measurement, the most…*" with "*At the end of the run, the most…*"

Line 2:" continuously at 10 ml min-1 flowing reference cell signals". Revise the order of words in the sentence.

We changed the sentence from

"*The values reported by the LI-7000 are Δ-signals which are basically the difference between the sample and the continuously at 10 mL min$^{-1}$ flowing reference cell signals.*"

to

"*The values reported by the LI-7000 are the difference between the signals of the sample and the reference cell, which is flushed continuously with a reference gas at a flow rate of 10 mL min$^{-1}$.*"

Line 16: C2 was measured twice? Should this be C2, C3?

This is a mistake, it was changed from "*C2*" to "*C1*".

Line 17: define what is "one run"

"One run" is when one or several cylinders are measured from their initial pressure down to pressure 1 (or 1.5 bar in the high flow setup).

We changed the sentence at page 4, line 17 from (first appearance of "run")

"*…which allowed measuring all sample cylinders in one run. With the new 16-port…*"

to

"*…which allowed measuring all sample cylinders from their initial pressure down to their final pressure in one single run. With the new 16-port…*"

Line 19: "secondary". This term may be replaced here as it was already used previously in the traceability hierarchy.

We replaced "*secondary*" with "*working*". For consistency reasons also at page 5, line 18 and deleted "*primary*" at page 3, line 28.

Line 26:"before a block of C2 was measured". As the word "block" is not so appropriate to a 5 minute average, the entire sentence is difficult to understand.

This whole section was changed, the sentence

*"In the low flow measurements with multiple samples, all samples were subsequently measured twice before a block of C2 was measured, and this was repeated three times, before another full calibration was done."*

reads now

*"In the low flow setup with multiple samples, the sequence cycled between the sample cylinders until two blocks of every sample cylinder were subsequently measured. Then a block of C1 was measured to catch the analyzers short term drift. This was repeated three times, before another full calibration was done."*

Line 31:"lasts" should be "lasted".

Done.

Line 28: "showed that they perform even slightly better." Sounds more like an opinion than an observation. Values should be provided, or the statement can simply be removed as it does not bring additional value.

The sentence was removed.

Line 15:"as soon as the cylinder reached 30 bar". Omission of the word "pressure", to be included.

Done.

Equation (2): using the symbol of the molecule for its mole fraction is confusing when it is used in the text. It is recommended to follow IUPAC terminology.

This is how Leuenberger et al., (2015) give the formula in their publication. However we changed "$CO_{2,meas}$" to "$X_{CO2,meas}$", "$CO_{2,ini}$" to "$X_{CO2,ini}$" and "$CO_{2,ad}$" to "$X_{CO2,ad}$" in the whole manuscript. For consistency, "$CO_{2ad,lf}$" was changed to "$X_{CO2ad,lf}$" as well.

Line 23:"amount density". This should be "amount concentration".

Done.

Line 24: "function of $CO_2$ only". This illustrates the problem of using the molecule's name instead of a symbol for its mole fraction.

We changed the sentence

*"In this approach θ is a function of $CO_2$ only, not of total gas pressure."*

to

*"In this approach, θ is a function of the amount of $CO_2$ only, not of total gas pressure."*

Line 2:"available". The term can be misleading, as it could indicate non occupied sites. May be replaced with "total number of sites".

We changed it from

*"...with θ the fraction available wall space that is occupied..."*

to

**"...**with θ the fraction of total number of available sites that are occupied..."*

Line 2:"amount of sites, expressed in moles". The mole is limited to an amount of molecules, atoms, ions, electrons, or other particles. This may be replaced with "maximum amount of adsorbed molecules", which also corresponds to the total number of sites.

Done.

Line 23 to 25: revise the sentence, maybe using two distinct sentences.

We changed

*"A fit following Langmuir's adsorption/desorption model was calculated for each measurement. The fit functions were used to calculate the average $CO_2$ enrichment with decreasing pressure using the pressure measurements, it is $0.089 \pm 0.013$ μmol mol$^{-1}$ (Fig.6 a), the given error corresponds to the standard deviation (1-sigma) of the average. Since each cylinder started with a different pressure, the averages could be not entirely comparable. However, if the enrichment is calculated over the same pressure span of 150 to 1 bar, the result is again $0.090 \pm 0.009$ μmol mol$^{-1}$, which is the same within the given uncertainty."*

to

*"A fit following Langmuir's adsorption/desorption model was calculated for each cylinder measurement and used to estimate the average $CO_2$ enrichment with decreasing pressure. Using the actual pressure measurements, the average $CO_2$ enrichment is $0.089 \pm 0.013$ μmol mol$^{-1}$ (Fig.6 a), the given error corresponds to the standard deviation (1-sigma) of individual cylinder drainings. However, values for the $CO_2$ enrichment of the individual cylinder measurements might be not entirely comparable, since each cylinder had a different initial pressure. Therefore, we calculated the $CO_2$ enrichment for each cylinder measurement using the same pressure span of 150 to 1 bar. This results in an average $CO_2$ enrichment of $0.090 \pm 0.009$ μmol mol$^{-1}$, which is the same within the given uncertainty. The variation of the enrichment was very low, indicating that the $CO_2$ enrichment with decreasing pressure is not cylinder dependent. The two SGS cylinders do not show a significantly different behavior, the form of the $CO_2$ enrichment with decreasing pressure as well as the amount is the same as for the normal cylinders within the given uncertainty (Fig. 6 b)."*

Line 24:"standard error". A better term would be "standard deviation", as used later on in the text.

True, the value given corresponds to the standard deviation, not to the standard error. We changed it accordingly.

Line 15: revise the last part of the sentence. It seems that some words are missing.

We changed

"*The average fractionation factor α is 0.99993 ± 0.00002, which means $CO_2$ depletion in the outflowing gas.*"

to

"*The average fractionation factor α is 0.99993 ± 0.00002, which corresponds to a $CO_2$ depletion in the outflowing gas.*"

Done.

Line 13: "are the virtually". Remove "the" and consider replacing the word "virtually", which seems more appropriate for a talk than for a paper.

We changed the sentence from

"*The temperature measurements of the two cylinders are the virtually the same (Fig. 16 a).*"

to

"*The temperature measurements of the two cylinders are in good agreement (Fig. 16 a).*"

Line 15: "By using a ….the number becomes even smaller". Consider revising the sentence, first because the fraction of occupied sites goes from 13% to 37%, secondly because the second calculation seems to be an alternative one, so that two models are compared.

The sentence

"*By using a very simplified geometrical approach, this number becomes even smaller.*"

was changed to

"*Using a very simplified geometrical approach results in a higher estimate of the occupied wall spaces.*"

Line 17: "the number of molecules…Avogadro number". Is it really needed to state what is the Avogadro number?

"…(Avogadro's number)…" has been removed and we changed the line to "…$N_A = 6.022 \cdot 10^{23}$ mol$^{-1}$, a pressure of $P = 150$ bar, a volume of $V_{cyl} = 29.5$ l, a temperature of $T = 293.15$ K…"

Line 24:"a second conclusion": what was the first conclusion?

We changed the sentence (page 15, line 5)

"*That suggests that the observed $CO_2$ enrichment may be universal for this type of aluminum cylinder.*"

to

"*Therefore the conclusion can be drawn that the observed $CO_2$ enrichment for ambient level $CO_2$-in-dry-air mixtures stored in this type of aluminum cylinder is universal.*"

Lines 9 to 12: consider splitting the sentence. This is currently difficult to understand.

We changed the sentence from

"*If it is laid down from vertical, the vanishing gradient along the cylinder and the emerging temperature difference between the now lower and upper side are proof of that the jump in the $CO_2$ mole fraction when a cylinder is laid down is in accordance with the aforementioned thermal diffusion fractionation that $CO_2$ gets enriched in cool air that accumulates at the bottom of the cylinder.*"

to

"*If a cylinder's orientation is changed from a vertical to a horizontal position, the temperature gradient initially remains the same along the cylinder. A temperature difference between the now lower and upper cylinder wall starts to build up. The conservation of the gradient's vertical orientation proves that the jump in the $CO_2$ mole fraction that occurs when a cylinders orientation is changed is in accordance with the aforementioned thermal diffusion fractionation, where $CO_2$ gets enriched in cool air that accumulates at the bottom of the cylinder.*"

Line 12 and 13: "to describe the $CO_2$ enrichment". Consider removing one of the appearances of this part of the sentence.

Referee#1 suggested to rewrite the conclusions, this statement reads now:

"*In low flow settings (0.3 L min$^{-1}$), the Langmuir adsorption/desorption model using averaged coefficients is sufficient to describe the $CO_2$ enrichment.*"

Figures 6, 8, 12: the y-axis is a difference, not the $CO_2$ mole fraction. From the legend it seems that the first value was subtracted from the set, but then why is the first point not at zero?

(A similar answer was given to Referee#1)

The y-axis corresponds to the $\Delta CO_2$ and was changed accordingly. For this figures, the term $X_{CO2,ini}$ ($CO_{2,ini}$ in the unrevised manuscript) was subtracted from the Langmuir-equation and the remainder with the corresponding coefficients was plotted. Because $X_{CO2,ini}$ corresponds to the $CO_2$ mole fraction before adsorption to the walls occurs, it is slightly higher than the average $CO_2$ mole fraction at the beginning of the measurements. This results in negative $\Delta CO_2$ values in the beginning of the measurements. For clarity we shifted the values by adding $X_{CO2,ad}$, this shifts the fits upwards so that the initial value is exactly 0. This was done for Fig 6, 8, and 12. In all three captions we changed

"...*the corresponding $X_{CO2,ini}$ was subtracted...*"

to

"...*the corresponding ($X_{CO2,ini} – X_{CO2,ad}$) was subtracted...*",

accordingly.